# Beyond Unidirectional Bias: Reciprocal Perspective Calibration in Scene Graph Generation

Haifeng Zhao [* 1]  Wenbo Zhao [* 1]  Xuemei Luo [2]  Lei-Lei Ma [3 1]  Dengdi Sun [4 1]

## Abstract

Scene Graph Generation (SGG) paradigms predominantly model relationships as static, unidirectional mappings ($s \rightarrow o$), effectively treating objects as passive recipients of actions. This formulation suffers from an inherent *unidirectional bias*, violating the physical reality that visual interactions are intrinsically reciprocal. Consequently, existing models often fail to maintain logical self-consistency when the reasoning anchor shifts from the subject to the object. To rectify this cognitive deficiency, we establish the Mutual-Perspective Inverse Relations (MPIR) principle, positing that a robust visual representation must satisfy logical consistency across dual perspectives. Guided by this principle, we propose the **Reciprocal Perspective Calibration (RPC)** framework, a model-agnostic framework that operationalizes MPIR via a novel Adaptive Inverse-Relation Augmentation (AIRA) strategy. Furthermore, we introduce Hypernym-Guided Prompts (HGP) to bridge the gap between semantic context and computational efficiency in vision-language models, enabling precise modeling of inverse relations. Extensive experiments demonstrate that RPC not only achieves competitive performance on standard benchmarks but also significantly enhances the model's capability to understand inverse relations, as verified by a new inverse consistency evaluation protocol, demonstrating the cognitive robustness of our method.

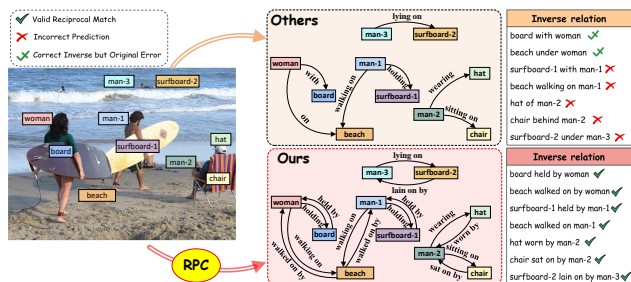

*Figure 1.* Illustration of unidirectional bias versus perspective invariance. Existing models fail to generalize inverse inputs, yielding contradictory predictions. Our approach captures the reciprocity of interactions, correctly inferring inverse relations. "Inverse relation" denotes predictions after swapping subject and object.

## 1. Introduction

Scene Graph Generation (SGG) aims to parse visual scenes into structured semantic representations. Formally, a visual scene is represented as a graph $\mathcal{G} = (\mathcal{V}, \mathcal{E})$, where objects are modeled as nodes ($\mathcal{V}$: node set) and their pairwise relationships as directed edges ($\mathcal{E}$: edge set) (Krishna et al., 2017). Each edge corresponds to a semantic triplet $\tau = \langle s, p, o \rangle$, consisting of subject and object nodes $s, o \in \mathcal{V}$ and a predicate category $p \in \mathcal{P}$.

This structured output bridges the gap between low-level visual perception and high-level reasoning, serving as a cornerstone for downstream tasks such as visual question answering, image retrieval, and image captioning.

Following the standard technical paradigm, most existing methods rely on object detectors to infer inter-object relationships, which are typically modeled as a strictly unidirectional mapping from the subject to the object ($s \rightarrow o$) via a predicted predicate $p$ (He et al., 2022). While this formulation conforms to the active-voice conventions of natural language, it solely anchors the description on the subject, inevitably introducing an intrinsic **unidirectional bias**. In essence, a predicate $p$ is merely a subjective linguistic description, whereas the underlying visual interaction—the relationship itself—is an objective physical fact that inherently possesses **reciprocity** rather than unidirectionality. As illustrated in Fig. 1, the visual fact $\langle man, holding, surfboard \rangle$ logically entails its reciprocal state $\langle surfboard, held\,by, man \rangle$.

---

[*]Equal contribution  [1]Anhui Provincial Key Laboratory of Multimodal Cognitive Computation, School of Computer Science and Technology, Anhui University, Hefei, Anhui, China [2]School of Computer Science & Instituteof Big Data, Fudan University, Shanghai, China [3]School of Artificial Intelligence Engineering, Hefei Institute of Technology, Hefei, Anhui, China [4]School of Artificial Intelligence, Anhui University, Hefei, Anhui, China. Correspondence to: Dengdi Sun (primary contact) <sundengdi@163.com>, Lei-Lei Ma <xiaoleilei1990@gmail.com>.

*Proceedings of the 43rd International Conference on Machine Learning*, Seoul, South Korea. PMLR 306, 2026. Copyright 2026 by the author(s).

However, confronting the reciprocal nature of interactions, existing paradigms exhibit critical deficiencies in both data and modeling. ❶ Widely-used SGG datasets suffer from severe intrinsic directional bias, where specific object categories are stereotypically labeled as fixed subjects or objects, and predicate annotations strictly follow the $s \rightarrow o$ direction (e.g., anchoring predominantly on *man*/*woman* in active voice). ❷ Existing models are confined to learning within a truncated unidirectional channel, forcing them to describe interactions solely from the subject's perspective while neglecting the object's potential as a descriptive anchor. Consequently, when the reasoning perspective shifts (anchoring on *board* instead of *person*), models often yield semantically collapsed or contradictory predictions (e.g., in Fig. 1, other models misclassifying $\langle beach, walking\ on, man \rangle$ from the object's view). Since a single image typically contains multiple triplets representing diverse relationships, the resulting scene graph becomes increasingly susceptible to the cumulative effects of such unidirectional bias.

These dual issues above contradict the intrinsic nature of relationships, culminating in a persistent **unidirectional bias** in SGG that hinders the efficacy of downstream tasks. Therefore, a model that truly comprehends the essence of interactions should adhere to the **perspective invariance** of physical facts, meaning that regardless of whether the description anchor is cast on the subject or the object, the underlying visual fact must remain invariant.

To bridge this cognitive gap, we establish the logical foundation of the **Mutual-Perspective Inverse Relations (MPIR)** principle. MPIR posits that a valid visual representation must satisfy logical consistency across dual perspectives. To achieve this, we formalize the concept of inverse relations. For every standard triplet $\tau$, we define a logically equivalent inverse triplet $\tau^{-1} = \langle o, p^{-1}, s \rangle$, where $p^{-1}$ is the inverse predicate.

Guided by this principle, we propose the **Reciprocal Perspective Calibration (RPC)** framework. It instantiates the MPIR constraints via a logic-driven Adaptive Inverse-Relation Augmentation (AIRA) strategy. Recognizing the disparities across predicate categories, AIRA disentangles the training protocol to deploy two tailored strategies—Additive Injection and Stochastic Replacement. Notably, RPC is designed as a **plug-and-play**, **model-agnostic** module. It can be seamlessly integrated into various existing SGG architectures without altering their intrinsic backbone designs, functioning as a universal calibration mechanism that rectifies the deficiency in inverse relation recognition and compensates for the perspective asymmetry that fundamentally leads to unidirectional bias.

However, effectively deploying RPC requires a model capable of understanding the semantic logic of inverse relations. While Vision-Language Models such as CLIP (Rad-ford et al., 2021) offer the necessary semantic priors and zero-shot capabilities, they face a dilemma in prompt engineering, where conventional prompt templates discard subject-object information, thereby lacking the context to distinguish directionality, whereas full-triplet enumeration incurs a combinatorial explosion of the search space scaling to $\mathcal{O}(|\mathcal{V}|^2 \times |\mathcal{P}|)$. To circumvent this dilemma, we introduce **Hypernym-Guided Prompts (HGP)**. By leveraging Large Language Models (LLM) to map fine-grained entities to semantic hypernyms (e.g., mapping *boy* to *person*), HGP abstracts the fine-grained entity set $\mathcal{V}$ into a compact hypernym set $\mathcal{H}$ (satisfying $|\mathcal{H}| \ll |\mathcal{V}|$). This abstraction effectively bridges the gap between semantic richness and computational efficiency, reducing the prompt search space while retaining vital subject-object cues. Consequently, HGP empowers CLIP to accurately discern the inverted perspectives required by RPC without prohibitive computational costs, establishing a robust visual-semantic verification mechanism that effectively resolves directional ambiguity.

Our main contributions are summarized as follows:

• We identify the cognitive limitation of unidirectional bias in SGG and establish MPIR as a principle, advocating that visual facts are intrinsically perspective-invariant, necessitating logical consistency under perspective shifts.

• We propose the RPC framework, a plug-and-play module that instantiates MPIR via AIRA strategy. By selectively applying additive or replacement protocols, it seamlessly rectifies the model's deficiency in inverse relation recognition across diverse architectures, enhancing adaptability to perspective shifts.

• We design HGP to adapt the RPC framework to CLIP-based architectures. HGP resolves the prompt complexity trade-off, enabling precise inverse relation modeling via semantic abstraction.

• Experiments demonstrate that, even disregarding inverse relation capabilities, RPC mitigates long-tail distribution and information imbalance. Furthermore, we design a novel inverse consistency evaluation protocol based on original settings to assess the model's resilience to unidirectional bias and capacity for inverse relation recognition.

## 2. Related Work

**Evolution of SGG Models.** SGG has evolved from early contextual message-passing mechanisms (Xu et al., 2017; Zellers et al., 2018; Tang et al., 2019) to recent global attention architectures (Li et al., 2022) and prototype-based semantic regularization (Zheng et al., 2023). More recently, Kim et al. (2024a) integrates global co-occurrence knowledge with learnable TF-IDF priors to further mitigate the long-tail distribution challenge. However, these paradigms

remain anchored on subject-centric unidirectional mappings ($s \rightarrow o$), neglecting the intrinsic reciprocity of physical interactions and failing to maintain consistency from the inverse perspective.

**Prompt Learning in SGG.** The integration of CLIP (Radford et al., 2021) has shifted SGG toward semantic alignment. While pioneering works in open-vocabulary SGG (He et al., 2022; Yu et al., 2023) and standard SGG (Zhu et al., 2024) have introduced learnable soft prompts, a critical trade-off persists. As highlighted by Zhu et al. (2024), concise templates suffer from semantic ambiguity, while full-triplet enumeration triggers a combinatorial explosion. Our HGP resolves this dilemma by leveraging LLM to inject coarse-grained hypernyms into prompts, bridging the gap between semantic clarity and computational feasibility.

**Inverse Relations in Knowledge Graph.** While Knowledge Graph (KG) embeddings (Kazemi & Poole, 2018; Sun et al., 2019; Chao et al., 2021) explicitly enforce logical reciprocity ($r(x, y) \iff r^{-1}(y, x)$), SGG paradigms typically treat predicates as isolated labels, leading to contradictory predictions upon viewpoint shifts. We are the first to introduce such strict logical consistency into SGG, bridging statistical fitting with physically grounded reasoning.

## 3. Motivation and Principle

### 3.1. The Intrinsic Unidirectional Bias in SGG

Despite architectural advancements, current SGG paradigms remain fundamentally built upon unidirectional mapping. We argue this profound bias stems from two sources: inherent flaws in data annotation protocols, and a systematic neglect of "relational reciprocity" in existing models.

**Asymmetric Annotation and Anthropocentric Bias.** While visual interactions are intrinsically reciprocal, standard SGG datasets suffer from severe annotation asymmetry. This manifests as a **subject selection preference**—driven by perceptual salience and anthropocentric perspectives, annotators habitually anchor descriptions on specific categories (typically large, animate, or dynamic entities like *person*) as subjects, relegating static entities to subordinate roles. Furthermore, annotations adhere to the active-voice convention, recording only $\langle s, p, o \rangle$ while neglecting the reciprocal expression flowing from the object to the subject. This bias results in a truncated training distribution, depriving models of exposure to the full spectrum of relational perspectives.

**Topological Sparsity and Reciprocal Collapse.** Prevailing annotation scarcity results in highly sparse graph structures, where the rich bi-directional connectivity inherent to real-world interactions is largely absent in existing datasets. This sparsity not only constrains reasoning capabilities for downstream tasks but critically impedes the perception of rela-

tionships as "bi-directionally locked" visual facts. Current modeling paradigms further aggravate this issue by treating predicates merely as unidirectional labels for statistical fitting, ignoring the perspective invariance of relational properties. Consequently, during inference, these models often yield semantically contradictory or collapsed predictions when subject-object roles are swapped.

**Benefits of Incorporating Inverse Relations.** Introducing the concept of inverse relations serves two critical purposes. ❶ **Logical Regularization**. It compels the model to comprehend the essence of interactions beyond one-way statistical correlation, effectively performing a semantic densification of the graph. ❷ **Downstream Utility**. For tasks such as Visual Question Answering and Image Retrieval, a reciprocally complete scene graph provides a more robust and information-rich representation, allowing for more flexible querying from arbitrary entity anchors. For instance, VQA tasks frequently feature highly dynamic question structures where the subject-object anchor shifts unpredictably; a unidirectional graph limits the VQA model's ability to answer object-centric queries accurately.

### 3.2. Inverse Relation Principle

#### 3.2.1. PRELIMINARIES

Following the standard SGG paradigm, given an image $\mathcal{I}$, the goal is to generate a scene graph $\mathcal{G} = (\mathcal{V}, \mathcal{E})$, where $\mathcal{V} = \{v_i\}_{i=1}^N$ and $\mathcal{E} = \{r_{ij} | v_i, v_j \in \mathcal{V}\}$. Each node $v_i$ is characterized by a bounding box $b_i \in \mathbb{R}^4$ and a semantic class label $c_i \in \mathcal{C}_{obj}$. Each edge $r_{ij}$ is formalized as a labeled triplet $\tau = \langle s, p, o \rangle$, where $s$ (subject) and $o$ (object) correspond to entities $v_i$ and $v_j$, and $p \in \mathcal{C}_{pred}$ represents the predicate category. Although the standard directed graph formulation aligns with linguistic active voice conventions, it introduces the aforementioned implicit directional entanglement. Consequently, models become overly reliant on fixed annotation orders, failing to capture the holistic structural semantics embedded within the reciprocal perspective.

#### 3.2.2. MUTUAL-PERSPECTIVE INVERSE RELATIONS

To remedy the aforementioned logical deficiency, we argue for a rigorous distinction between the linguistic unidirectionality of a predicate and the intrinsic bidirectionality of the underlying visual relation, and formally introduce the **MPIR** principle. Grounded in physical reality, we posit that every reversible interaction manifests as a visual fact, invariant to the observer's perspective. Consequently, the primal relation and its inverse must logically satisfy the same intrinsic truth. We formalize this visual fact consistency as:

$$\mathcal{R} \models \langle s, p, o \rangle \iff \mathcal{R} \models \langle o, p^{-1}, s \rangle, \quad (1)$$

where $\mathcal{R}$ denotes the physical visual reality (depicted by image $\mathcal{I}$) and $\models$ represents semantic satisfaction.

To operationalize this logical axiom, we define the inverse triplet structure and based on it, establish a rigorous bijective mapping mechanism. By shifting the observational anchor from the subject to the object, this mechanism derives a new triplet $\langle o, p', s \rangle$ describing the identical visual fact, where $p'$ is the inverse predicate of $p$, denoted as $p^{-1}$. Together, $p$ and $p^{-1}$ constitute a *reciprocal predicate pair*, describing the same visual fact under inverted observer perspectives:

$$\tau = \langle s, p, o \rangle \underset{\Phi^{-1}}{\overset{\Phi}{\rightleftarrows}} \tau^{-1} = \langle o, p^{-1}, s \rangle. \qquad (2)$$

Here, $\Phi$ and $\Phi^{-1}$ form the bijective mapping: $\Phi$ transforms the primal triplet into its inverse form, while $\Phi^{-1}$ restores it, ensuring strict logical reversibility. When the subject and object entities are swapped, the perspective anchor shifts from the original subject to the object, necessitating the emergence of the new predicate $p^{-1}$ to maintain the original semantic truth value of $\mathcal{R}$. This renders the generated inverse triplet logically equivalent to the primal one.

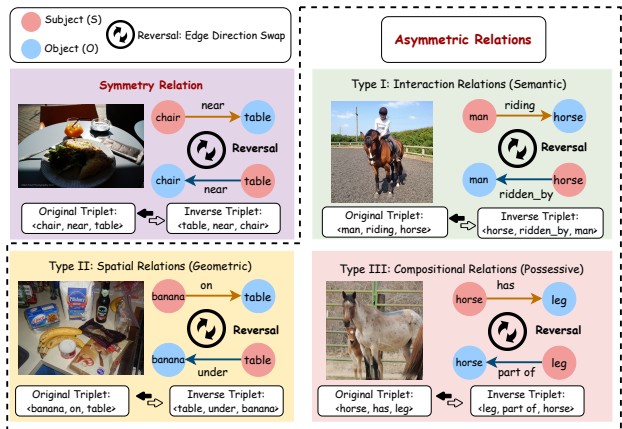

*Figure 2.* Predicates are classified as **Symmetric** (invariant) or **Asymmetric**, where the latter comprises SEMANTIC, GEOMETRIC, and POSSESSIVE subtypes, each requiring specific inverse mappings to preserve the visual fact.

### 3.2.3. CATEGORIZATION OF INVERSE RELATIONS

To facilitate adaptive learning, we categorize inverse relations based on their logical properties and transformation mechanisms (Fig. 2). Fundamentally, predicates exhibit two distinct logical behaviors. The first is **Symmetry** (Self-Inverse), where the predicate remains identical upon inversion (i.e., $p^{-1} = p$, implying $\forall x, y : r(x, y) \iff r(y, x)$). The second is **Asymmetry**, where the inverse requires a specific semantic reversal (i.e., $p^{-1} \neq p$) to maintain logical consistency. While symmetric relations (e.g., *near*, *and*) are trivial, the majority of semantically rich visual interactions are inherently Asymmetric. Consequently, simply swapping entities without transforming the predicate yields logical fallacies. Following established conventions in SGG literature, we partition the predicate space $\mathcal{P}$ into three super-categories

$\mathcal{SC} = \{\text{SEMANTIC}, \text{GEOMETRIC}, \text{POSSESSIVE}\}$. They require explicit, type-specific semantic mappings to preserve the underlying visual fact.

**Type I: Semantic Relations (SE)** characterize dynamic actions where an agent (subject) acts upon a patient (object). The inverse mapping $\Phi$ typically manifests through linguistic **active-passive voice alternation**, e.g.,
• $\langle man, riding, horse \rangle \leftrightarrow \langle horse, ridden\ by, man \rangle$.

**Type II: Geometric Relations (GE)** describe relative positions in physical space. The inverse predicate is derived via spatial reversal logic, for instance,
• $\langle cup, on, table \rangle \leftrightarrow \langle table, under, cup \rangle$.

**Type III: Possessive Relations (PO)** describe stable structural dependencies, primarily encompassing part-whole hierarchies and ownership. The transformation entails a reciprocal interchange between coupled roles, specifically pairing "WHOLE" with "PART", and "OWNER" with "POSSESSION":
• $\langle person, has, hand \rangle \leftrightarrow \langle hand, part\ of, person \rangle$,
• $\langle phone, belonging\ to, man \rangle \leftrightarrow \langle man, own, phone \rangle$.
This structural inversion ensures the stability of possession semantics across perspectives.

It is worth noting that the principle of reciprocity is not merely a statistical heuristic but a logical axiom validated in structured knowledge systems, such as KG. The established success of modeling logical symmetry in these non-visual domains provides strong **cross-disciplinary corroboration** for our MPIR principle. It suggests that ensuring logical consistency is not an optional constraint but a fundamental prerequisite for robust reasoning, further validating the necessity of grounding SGG in physical reality.

## 4. Methodology

### 4.1. Reciprocal Perspective Calibration (RPC)

While the MPIR principle establishes the logical necessity of visual duality, standard training pipelines remain constrained by the biases inherent in static datasets. To bridge this gap between theoretical truth and empirical learning, we propose the RPC framework. It is defined as a holistic calibration system that operationalizes the logical constraints of MPIR through a data-centric AIRA strategy. By directly translating the abstract MPIR into tangible training signals, RPC functions as a plug-and-play module that aligns visual features with logical reciprocity.

### 4.1.1. ADAPTIVE INVERSE-RELATION AUGMENTATION

Generally, predicate categories exhibit distinct disparities. In the long-tailed distribution, divided into HEAD (H), BODY (B), and TAIL (T), the frequency of H vastly outweighs T. From the super-category perspective, GE predicates (Type

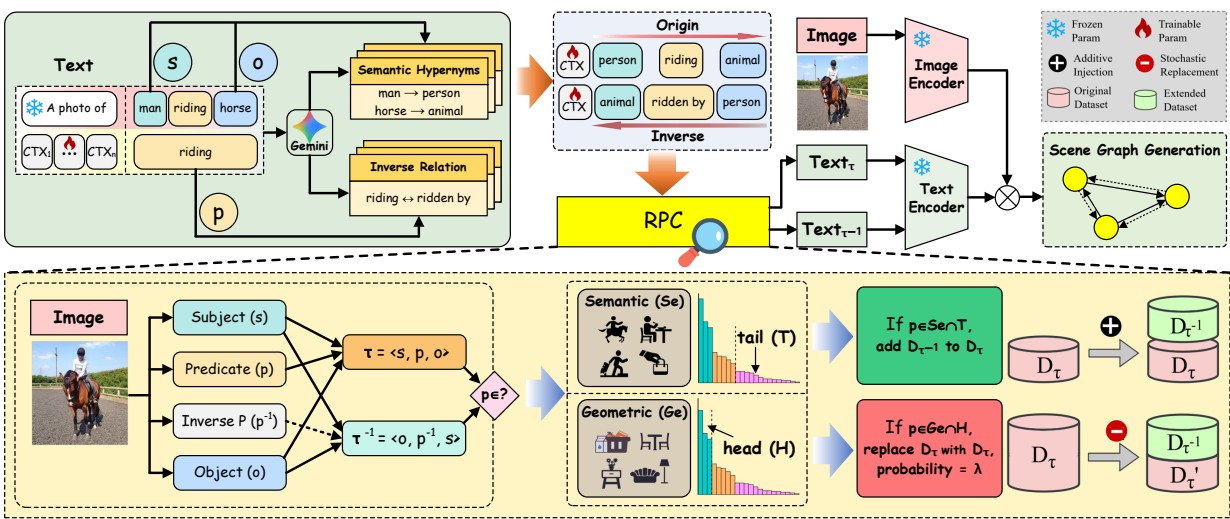

*Figure 3.* Overview of the RPC framework. **Top:** The pipeline utilizes HGP to inject semantic hypernyms and generate reciprocal triplet pairs via the **MPIR** principle, aligning text embeddings with visual features in the CLIP latent space. **Bottom: AIRA** strategy applies Additive Injection for SEMANTIC or TAIL predicates to enrich distribution sparsity, and **Stochastic Replacement** (with probability $\lambda$) for GEOMETRIC or HEAD predicates to counteract positional bias. By integrating these dual-perspective constraints, the framework effectively calibrates the cross-modal alignment to produce topologically robust and unbiased scene graphs.

II) establish strong but trivial priors, whereas SE predicates (Type I) are semantically rich yet critically sparse. Consequently, uniformly applying MPIR is inappropriate. We thus disentangle these factors to deploy two targeted protocols, as illustrated in Fig. 3.

❶ **Additive Injection.** This strategy is deployed for predicates characterized by low frequency or high semantic density (Type I). For these relations, the primary barrier is the scarcity of supervision signals (e.g., missing $\langle snow, covering, tree \rangle$ despite the presence of $\langle tree, covered\ in, snow \rangle$). We employ an Additive Injection strategy (hereafter denoted as **Add**) to explicitly maximize semantic exposure. For every valid triplet $\tau$ in these categories, we generate its inverse $\tau^{-1}$ via the mapping $\Phi$ and inject it into the training set:

$$\tau^{-1} = \Phi(\tau), \qquad \mathcal{D}_{train} \leftarrow \mathcal{D}_{train} \cup \{\tau^{-1}\}. \quad (3)$$

Here, $\mathcal{D}_{train}$ represents the training dataset of visual relationship triplets. This functions as a semantic counterbalancing mechanism, compelling the model to learn complex reciprocal logic often lost in standard training.

❷ **Stochastic Replacement.** It is applied to predicates exhibiting high frequency or geometric redundancy (Type II). These relations typically saturate the dataset, leading models to overfit to positional biases (e.g., defaulting to *on*, which hinders the recognition of fine-grained triplets like $\langle bike, parked\ on, street \rangle$) rather than visual features. For such cases, expanding the dataset is redundant. Instead, we utilize a Stochastic Replacement strategy (denoted as **Rep**) to enforce robustness without increasing the training scale. Specifically, we stochastically perturb the triplet $\tau$ using its

inverse projection $\Phi(\tau)$ to generate a training instance $\tilde{\tau}$ with a swapping probability $\lambda$:

$$\tilde{\tau} = \delta \cdot \tau^{-1} + (1 - \delta) \cdot \tau, \quad \text{where } \delta \sim \text{Bernoulli}(\lambda). \quad (4)$$

Here, $\delta \in \{0, 1\}$ acts as a stochastic binary selector sampled from a Bernoulli distribution with parameter $\lambda$. This mutually exclusive exposure to either $\tau^{-1}$ (probability $\lambda$) or $\tau$ (probability $1 - \lambda$) functions as a perspective-invariant regularizer, disrupting reliance on fixed ordering without exacerbating the dominance of H or Ge categories.

### 4.1.2. VOCABULARY-AWARE ADAPTATION

The execution of the aforementioned protocols is contingent on architectural constraints. For open-set models (e.g., CLIP), the mapping $\Phi(\tau)$ is applied unrestrictedly to introduce novel inverse predicates. However, for closed-set baselines restricted by a fixed vocabulary $\mathcal{C}$, we impose a validity filter, such that the generated inverse triplet $\tau^{-1}$ (in Add) or $\tilde{\tau}$ (in Rep) is utilized if and only if the inverse predicate $p^{-1} \in \mathcal{C}$. This constraint ensures that AIRA precludes the introduction of Out-of-Distribution (OOD) noise to the closed-set classifiers. Alternative strategies for handling OOD predicates are detailed in Appendix B.

In conclusion, RPC leverages adaptive augmentation to inject reciprocal logic as a plug-and-play module, effectively neutralizing unidirectional bias. Crucially, strictly operating as a data-centric calibration, it is architecturally agnostic to underlying backbones or graph networks, thereby preserving the integrity of standard SGG pipelines.

## 4.2. Hypernym-Guided Prompt (HGP) Learning

Although our RPC framework functions theoretically as a model-agnostic augmentation strategy, practical deployment is hindered by the closed-set constraint of standard SGG. Since these models operate on fixed vocabularies, the direct injection of inverse relations as labels precipitates OOD failures, as classifiers lack semantic grounding to generalize logical inversions. To bridge this gap, we instantiate RPC within an open-set model (e.g., CLIP). Leveraging its pre-trained semantic space allows recognizing inverse interactions absent from training. However, adapting CLIP for SGG entails a unique challenge in constructing prompts that are both contextually precise and computationally efficient. To end this, we introduce HGP.

### 4.2.1. THE DILEMMA OF PROMPTING IN SGG

Although prompt tuning has succeeded in object recognition, its adaptation to SGG faces a unique context-injection dilemma. ❶ **Combinatorial Explosion**. Ideally, prompts should include specific entity labels (e.g., "a photo of [SUB] [PRED] [OBJ]", where placeholders denote the textual class names of the subject, predicate, and object)) to provide full context. However, this leads to an intractable search space. With $N_e$ entity classes and $N_p$ predicates, the triplet combinations scale to $\mathcal{O}(N_e^2 N_p)$, making optimization computationally prohibitive. ❷ **Contextual Loss**. Conversely, standard templates like "a photo of [PRED]" or prompts optimized with learnable vectors ($[\mathbf{U}]_m[\text{PRED}]$) avoid the combinatorial explosion but discard the semantic information of the subject and object. This is detrimental, as visual relations are inherently coupled with the entities involved.

### 4.2.2. SEMANTIC HYPERNYM MAPPING VIA LLM

To reconcile the conflict between semantic precision and computational feasibility, we introduce **semantic hypernyms** as a linguistic abstraction layer. Instead of using raw entity labels, we leverage the reasoning capabilities of LLM to map the fine-grained entity categories in the dataset into a compact set of $J$ coarse-grained hypernyms $\mathcal{H}$. The set $\mathcal{H}$ is represented as $\mathcal{H} = \{h_j\}_{j=1}^{J}$. Formally, we define a projection function $\Psi : \mathcal{V} \rightarrow \mathcal{H}$ that aggregates entities sharing functional or semantic attributes (e.g., $\Psi(man) = \Psi(boy) = person$). This projection reduces prompt complexity while retaining the critical semantic priors necessary for distinguishing relational contexts.

### 4.2.3. CONSTRUCTION OF HGP

Building upon these abstractions, we propose HGP. This architecture integrates learnable context vectors with fixed hypernym embeddings, offering an optimal trade-off between flexibility and semantic specificity.

Given a standard triplet $\tau = \langle s, p, o \rangle$, we construct the HGP $\mathcal{T}$ by concatenating a sequence of $M$ learnable context vectors $\mathbf{U} = \{\boldsymbol{u}_1, \ldots, \boldsymbol{u}_M\}$ with the embeddings of the corresponding hypernyms and predicate:

$$\mathcal{T}_{\text{org}} = \left[\boldsymbol{u}_1, \ldots, \boldsymbol{u}_M, \boldsymbol{e}_{\Psi(s)}, \boldsymbol{e}_p, \boldsymbol{e}_{\Psi(o)}\right], \qquad (5)$$

where $\boldsymbol{e}_{(\cdot)}$ denotes the fixed token embedding from CLIP, and $\Psi(\cdot)$ is the hypernym projection.

Significantly, HGP naturally extends to our RPC framework. While merely reversing the predicate label fails to capture the requisite perspective shift, HGP explicitly swaps the subject and object hypernyms, forcing the model to recognize the interaction from the inverted viewpoint. The prompt for the inverse triplet $\tau^{-1} = \langle o, p^{-1}, s \rangle$ is formulated as:

$$\mathcal{T}_{\text{inv}} = \left[\boldsymbol{u}_1, \ldots, \boldsymbol{u}_M, \boldsymbol{e}_{\Psi(o)}, \boldsymbol{e}_{p^{-1}}, \boldsymbol{e}_{\Psi(s)}\right]. \qquad (6)$$

By structurally aligning the textual prompt with the visual inverse transformation, HGP enforces synchronization between the text and image encoders within the shared latent space, maximizing the efficacy of perspective calibration.

## 4.3. LLM-Driven Knowledge Distillation

We align our predicate taxonomy with established structural definitions (Chen et al., 2024; Jiang et al., 2025), ensuring consistency with standard protocols. To derive the rigorous inverse mapping $\Phi : p \rightarrow p^{-1}$ and the semantic hypernym projection $\Psi : \mathcal{V} \rightarrow \mathcal{H}$, we distill the rich semantic priors inherent in LLM to instantiate these functions. Notably, rather than relying on unconstrained, open-ended LLM predictions—which are prone to semantic hallucinations and ambiguity—we leverage the structured taxonomy as few-shot anchors to strictly guide the generation process. This ensures an optimal trade-off between semantic precision and token efficiency, while perfectly aligning the distilled knowledge with our AIRA strategy. Furthermore, this constrained prompting paradigm renders our framework highly insensitive to the specific selection of the underlying LLM. As empirically validated in our ablation studies, deploying language models across diverse parameter scales and architectural capacities yields consistently stable results without significant performance discrepancies, thereby demonstrating the superior robustness and reproducibility of our methodology. Consistent with our model-agnostic philosophy, RPC introduces no auxiliary regularization terms, strictly adhering to the standard training paradigm on the calibrated dataset.

# 5. Experiments

## 5.1. Experimental Setup and Metrics

**Standard Metrics.** For evaluation, we report standard Recall@K (R@K) for general accuracy and Mean Recall@K

(mR@K) to assess performance on tail categories.

**Inverse Recall Metrics.** To quantify the model's capability in inverse relation recognition, we introduce a novel evaluation protocol. Based on the standard test set $\mathcal{D}_{test} = \{\langle s_i, p_i, o_i \rangle\}_{i=1}^N$, we construct an inverse query set $\mathcal{Q}_{inv} = \{\langle o_i, p_i^{-1}, s_i \rangle\}$, where the visual context remains identical, but the structural reasoning direction is reversed. The ground truth for each query corresponds to the logical inverse predicate $p_i^{-1}$. We define **Inverse Recall at K (IR@K)** as the proportion of samples where the target inverse predicate appears in the top-K predictions:

$$\text{IR@K} = \frac{1}{N} \sum_{i=1}^N \mathbb{I}\left(\text{rank}(p_i^{-1} \mid o_i, s_i) \leq K\right), \quad (7)$$

where $\mathbb{I}(\cdot)$ is the indicator function, and $\text{rank}(p \mid o, s)$ denotes the rank of predicate $p$ within the model's probability distribution $P(\cdot \mid o, s)$. Functioning as a stress test on the inverse manifold, IR@K effectively exposes the unidirectional bias inherent in standard evaluations.

To further mitigate class imbalance, we also define **Inverse Mean Recall (ImR@K)** by averaging IR@K across all predicate categories: $\text{ImR@K} = \frac{1}{|\mathcal{P}|} \sum_{c \in \mathcal{P}} \text{IR@K}_c$, where $\text{IR@K}_c$ denotes the inverse recall calculated specifically for the subset of triplets belonging to category $c$. This metric ensures that perspective invariance is maintained across the entire semantic spectrum, preventing the evaluation from being dominated by frequent head classes.

**Composite Metrics.** Following Zheng et al. (2023), we report Mean@K (M@K) to balance head/tail performance, defined as $\text{M@K} = (\text{R@K} + \text{mR@K})/2$. Similarly, to provide a holistic assessment of reciprocal consistency, we define Inverse Mean@K as $\text{IM@K} = (\text{IR@K} + \text{ImR@K})/2$.

Specifically, $\text{RPC}_{Rep}$ applies the Rep strategy ($\lambda = 0.5$) to relations in both HEAD (H) and GEOMETRIC (Ge) categories, while $\text{RPC}_{Add}$ targets predicates in the TAIL (T) and SEMANTIC (Se) intersection. Regarding notation, $\text{RPC}^*$ denotes the filtering strategy for closed-set models. Our evaluation consistently follows the PredCls protocol on the Visual Genome (VG) dataset.

### 5.2. Comparative Analysis

**Baselines.** To verify the generalization of RPC, we integrate it into three representative architectures (Motifs (Zellers et al., 2018), VCTree (Tang et al., 2019), PE-Net (Zheng et al., 2023)) and their CLIP-enhanced open-set variants ($\text{CLIP}_{M/V}$) (Zhu et al., 2024). We benchmark against diverse state-of-the-art methods, including TDE (Tang et al., 2020), IETrans (Zhang et al., 2022), ST-SGG (Kim et al., 2024b), MLLMP (Songju et al., 2025), VETO (Sudhakaran et al., 2023), GSGG (Zhu et al., 2025), and CooK (Kim et al., 2024a). (See Appendix A for details).

*Table 1.* Performance comparison on VG dataset (PredCl). $\text{RPC}_{Rep/Add}$: our calibration strategies. $\text{RPC}^*$: closed-set filtering. **Bold** and underline: the best and second-best results.

| | MODELS | R@50/100 | MR@50/100 | M@50/100 |
|---|---|---|---|---|
| *Motifs Series* | MOTIFS | 65.2 / 67.0 | 14.8 / 16.1 | 40.0 / 41.6 |
| | +$\text{RPC}^*_{REP}$ | 66.5 / 68.1 | 18.6 / 19.7 | 42.6 / 43.9 |
| | +$\text{RPC}^*_{ADD}$ | 67.1 / 68.5 | 18.5 / 19.4 | 42.8 / 44.0 |
| | $\text{CLIP}_M$ | 64.5 / 67.5 | 15.3 / 17.4 | 39.9 / 42.5 |
| | +$\text{RPC}_{REP}$+HGP | **68.6 / 69.1** | 19.3 / 21.3 | 43.9 / 45.2 |
| | +$\text{RPC}_{ADD}$+HGP | 67.9 / 68.5 | 21.0 / 22.8 | 44.4 / 45.7 |
| | TDE | 46.2 / 51.4 | 25.5 / 29.1 | 35.9 / 40.3 |
| | IETRANS | 54.7 / 56.7 | 30.9 / 33.6 | 42.8 / 45.2 |
| | ST-SGG | 63.4 / 65.4 | 22.4 / 24.1 | 42.9 / 44.8 |
| | MLLMP | 60.4 / 62.3 | 28.9 / 30.6 | 44.7 / 46.5 |
| *VCTree Series* | VCTREE | 65.9 / 67.8 | 15.7 / 16.8 | 40.8 / 42.3 |
| | +$\text{RPC}^*_{REP}$ | 65.9 / 67.4 | 17.4 / 18.7 | 41.6 / 43.0 |
| | +$\text{RPC}^*_{ADD}$ | 66.5 / 68.1 | 17.6 / 18.8 | 42.0 / 43.5 |
| | $\text{CLIP}_V$ | 66.4 / 67.9 | 16.0 / 17.6 | 41.2 / 42.7 |
| | +$\text{RPC}_{REP}$+HGP | 67.3 / 68.9 | 19.9 / 21.6 | 43.6 / 45.3 |
| | +$\text{RPC}_{ADD}$+HGP | 66.7 / 68.0 | 21.3 / 23.4 | 43.9 / 45.6 |
| | TDE | 47.2 / 51.6 | 25.4 / 28.7 | 36.3 / 40.2 |
| | IETRANS | 53.0 / 55.0 | 30.3 / 33.0 | 42.9 / 45.0 |
| | ST-SGG | 64.2 / 66.2 | 21.5 / 22.9 | 42.9 / 44.6 |
| | MLLMP | 60.8 / 62.9 | 29.1 / 30.6 | 45.0 / 46.8 |
| *Others* | VETO | 61.9 / 63.9 | 33.1 / 35.1 | 47.5 / 49.5 |
| | GSGG | 55.8 / 58.3 | **41.4 / 45.0** | 48.6 / **51.7** |
| | CooK | 60.4 / 62.3 | 35.4 / 37.2 | 47.9 / 49.8 |
| *PE-Net* | PE-NET | 60.0 / 61.9 | 35.4 / 37.0 | 47.7 / 49.5 |
| | +$\text{RPC}^*_{REP}$ | 57.4 / 59.7 | 40.0 / 42.2 | 48.7 / 51.0 |
| | +$\text{RPC}^*_{ADD}$ | 57.2 / 59.6 | 40.4 / 42.9 | **48.8** / 51.2 |

**Analysis.** As presented in Table 1, RPC demonstrates consistent effectiveness across diverse architectures, notably achieving these gains even under the standard evaluation protocol that disregards inverse relation capabilities. Since our core innovation focuses on relational calibration, we evaluate primarily under PredCls to isolate predicate recognition from compounding upstream detection errors. Performance gains in the more comprehensive evaluation protocols (i.e., SGCls and SGDet) are structurally driven by and derived from enhancements at this fundamental predicate level. Specifically, our experimental analysis reveals a hierarchical performance gain. **First**, regarding the three representative closed-set baselines, RPC substantially elevates their performance. **Then**, it is noteworthy that the magnitude of improvement is most pronounced in open-set CLIP variants. In this setting, the synergy between RPC and HGP yields a performance leap that far surpasses the gains observed in closed-set settings, as HGP effectively bridges the semantic gap to fully activate VLM priors. **Furthermore**, compared to other state-of-the-art models, our calibrated baselines exhibit remarkable competitiveness. Most notably, when integrated with the strong baseline PE-Net, the $\text{RPC}_{Add}$ strategy achieves performance highly com-

*Table 2.* Evaluation of Perspective Invariance on VG. We report standard metrics and our proposed Inverse metrics. Notably, while **PE-Net** shows the highest performance in the standard setting, **CLIP$_M$** demonstrates superior inverse relation recognition, validating the necessity of VLM priors for reciprocal reasoning.

| Model | R 50 | R 100 | MR 50 | MR 100 | M 50 | M 100 | IR 50 | IR 100 | ImR 50 | ImR 100 | MI 50 | MI 100 |
|---|---|---|---|---|---|---|---|---|---|---|---|---|
| Motifs | 65.2 | 67.0 | 14.8 | 16.1 | 40.0 | 41.6 | 5.5 | 6.3 | 2.1 | 2.6 | 3.8 | 4.4 |
| +RPC$^*_{Rep}$ | 66.5 | 68.1 | 18.6 | 19.7 | 42.6 | 43.9 | 6.9 | 8.5 | 4.6 | 5.8 | 5.8 | 7.1 |
| +RPC$^*_{Add}$ | 67.1 | 68.5 | 18.5 | 19.4 | 42.8 | 44.0 | 7.5 | 8.9 | 5.2 | 6.5 | 6.3 | 7.7 |
| VCTree | 65.9 | 67.8 | 15.7 | 16.8 | 40.8 | 42.3 | 6.9 | 7.9 | 5.2 | 6.0 | 6.1 | 7.0 |
| +RPC$^*_{Rep}$ | 65.9 | 67.4 | 17.4 | 18.7 | 41.6 | 43.0 | 20.8 | 22.1 | 5.4 | 6.1 | 13.1 | 14.2 |
| +RPC$^*_{Add}$ | 66.5 | 68.1 | 17.6 | 18.8 | 42.0 | 43.5 | 18.1 | 20.9 | 6.7 | 8.1 | 12.4 | 14.6 |
| CLIP$_M$ | 64.5 | 67.5 | 15.3 | 17.4 | 39.9 | 42.5 | 51.7 | 54.9 | 9.1 | 10.8 | 30.4 | 32.8 |
| +RPC$_{Rep}$+HGP | **68.6** | **69.1** | 19.3 | 21.3 | 43.9 | 45.2 | 55.4 | 59.0 | 10.2 | 11.6 | 32.8 | 35.3 |
| +RPC$_{Add}$+HGP | 67.9 | 68.4 | 21.0 | 22.8 | 44.4 | 45.7 | **57.6** | **60.7** | 10.8 | 12.1 | **34.2** | **36.4** |
| CLIP$_V$ | 66.4 | 67.9 | 16.0 | 17.6 | 41.2 | 42.7 | 50.8 | 53.9 | 9.3 | 11.1 | 30.0 | 32.5 |
| +RPC$_{Rep}$+HGP | 67.3 | 68.9 | 19.9 | 21.6 | 43.6 | 45.3 | 55.1 | 58.4 | 10.6 | 11.9 | 32.8 | 35.2 |
| +RPC$_{Add}$+HGP | 66.7 | 68.0 | 21.3 | 23.4 | 43.9 | 45.6 | 56.1 | 59.4 | 10.9 | 12.3 | 33.5 | 35.9 |
| PE-Net | 63.1 | 65.0 | 29.4 | 31.1 | 46.3 | 48.1 | 40.1 | 42.4 | 10.9 | 12.1 | 25.5 | 27.3 |
| +RPC$^*_{Rep}$ | 57.4 | 59.7 | 40.0 | 42.2 | 48.7 | 51.0 | 37.6 | 40.0 | 19.4 | 20.7 | 28.5 | 30.4 |
| +RPC$^*_{Add}$ | 57.2 | 59.6 | **40.4** | **42.9** | **48.8** | **51.2** | 35.9 | 38.4 | **23.5** | **24.9** | 29.8 | 31.7 |

petitive with the recent state-of-the-art work GSGG (Zhu et al., 2025), compellingly validating the efficacy of the RPC framework as a universal plug-and-play module.

*Table 3.* Overall ablation study on VG (PredCls). We evaluate HGP and RPC ($RPC_{Rep}$, $RPC_{Add}$) over the CLIP$_M$ baseline.

| Method | R@50/100 | mR@50/100 | M@50/100 |
|---|---|---|---|
| CLIP$_M$ | 64.5 / 67.5 | 15.3 / 17.4 | 39.9 / 42.5 |
| + RPC$_{Rep}$ | 66.7 / 68.1 | 18.2 / 19.6 | 42.5 / 43.9 |
| + RPC$_{Add}$ | 66.4 / 67.9 | 19.1 / 20.4 | 42.7 / 44.2 |
| + HGP | 65.7 / 67.7 | 20.4 / 21.9 | 43.1 / 44.7 |
| + HGP + RPC$_{Rep}$ | **68.6 / 69.1** | 19.3 / 21.3 | 43.9 / 45.2 |
| + HGP + RPC$_{Add}$ | 67.9 / 68.4 | **21.0 / 22.8** | **44.4 / 45.7** |

## 5.3. Reciprocal Consistency Evaluation

To assess robustness against perspective shifts, we evaluate models on the proposed Inverse Test Set. As shown in Table 2, standard models suffer from *inverse collapse* (e.g., Motifs drops to 4.4% MI@100) due to unidirectional statistical bias. In contrast, RPC consistently calibrates all backbones. Notably, while PE-Net leads in balanced metrics, **CLIP$_M$+RPC** achieves the highest perspective invariance (36.4% MI@100). This confirms that while structural priors (PE-Net) aid long-tail distribution, VLM semantic priors are decisive for ensuring logical symmetry in visual reasoning.

## 5.4. Ablation Studies

We conduct comprehensive ablation studies to verify the effectiveness of the modules in our proposed framework.

**Component Analysis.** It is worth noting that all ablation experiments in this section consistently employ CLIP$_M$ as

*Table 4.* Component ablation on VG (PredCls). We evaluate the individual and combined contributions of HGP and RPC strategies based on the CLIP$_M$ baseline.

| | Scope | R 50 | R 100 | mR 50 | mR 100 | M 50 | M 100 |
|---|---|---|---|---|---|---|---|
| *Rep* | All | 64.2 | 67.3 | 16.3 | 18.3 | 40.3 | 42.8 |
| | H | 66.7 | 68.1 | 18.1 | 19.1 | 42.4 | 43.6 |
| | Ge | 64.8 | 67.6 | 17.0 | 18.9 | 40.9 | 43.2 |
| | H + Ge | **66.7** | **68.1** | 18.2 | 19.6 | 42.5 | 43.9 |
| *Add* | All | 65.0 | 67.7 | 16.9 | 18.6 | 41.0 | 43.2 |
| | T | 64.4 | 66.8 | 18.1 | 19.8 | 41.2 | 43.3 |
| | Se | 65.5 | 67.1 | 19.6 | 21.0 | 42.6 | 44.0 |
| | T + Se | 66.4 | 67.9 | **19.1** | **20.4** | **42.7** | **44.2** |

the baseline model. On this basis, we evaluate the individual contributions of RPC and HGP. As shown in Table 3, results indicate distinct roles: RPC$_{Rep}$ primarily improves R@K, whereas RPC$_{Add}$ significantly boosts mR@K, offering superior overall gains between the two strategies.

**AIRA Strategy Optimization.** Focusing on the RPC module under the CLIP$_M$ setting, we analyze the target scope for two strategies. As presented in Table 4, for Add, targeting the intersection of T and S$_E$ relations yields the optimal overall balance (44.2% M@100), surpassing global application. Similarly, Rep proves most effective when restricted to relations that are both H and G$_E$, achieving the best trade-off between Recall and Mean Recall.

**LLM Selection for HGP.** We investigate the impact of the knowledge source by evaluating GPT-4o (Achiam et al., 2023), DeepSeek-V3 (Bi et al., 2024), Qwen-2.5 (Bai et al., 2023), and the Gemini-2.5 series (Team et al., 2024) (Table 5). While the Gemini series (specifically Gemini-2.5-pro) achieves the optimal results, the performance gap

*Table 5.* Ablation on LLM selection for HGP (PredCls). We compare variants utilizing different LLMs for semantic hypernym mapping. "+thinking" denotes refinement for task alignment.

| METHOD | R@50/100 | mR@50/100 | M@50/100 |
|---|---|---|---|
| GPT-4o | 65.8 / 67.5 | 18.5 / 20.1 | 42.1 / 43.8 |
| DeepSeek-V3 | 64.3 / 67.1 | 17.0 / 19.2 | 40.7 / 43.1 |
| Qwen-2.5 | 64.4 / 66.3 | 20.1 / 21.5 | 42.3 / 43.9 |
| Gemini-flash | 65.8 / 67.5 | 19.4 / 20.7 | 42.6 / 44.1 |
| Gemini-pro | 65.1 / 66.7 | **20.5** / 21.6 | 42.8 / 44.2 |
| *+thinking* | **65.7 / 67.7** | 20.4 / **21.9** | **43.1 / 44.7** |

across different LLMs is remarkably marginal, regardless of their disparate architectures or parameter scales (e.g., the comparable efficacy between DeepSeek-V3 and Gemini-2.5). This insensitivity to the choice of foundation models underscores the robustness and high reproducibility of our framework. Furthermore, directly querying LLMs for semantic hypernym generation is prone to hallucinations, logical fallacies, and format misalignment. To alleviate this, we design a "+thinking" strategy guided by the MPIR principles. By explicitly formulating MPIR constraints within the prompts, we regularize the output space and instruct the LLM to perform iterative self-verification. Notably, integrating this strategy with Gemini-2.5-pro yields the peak performance of 44.7% on M@100, concretely demonstrating the effectiveness of such logical constraints. This confirms that, without relying on exhaustive manual rules, LLMs can effectively generate task-aligned semantics under simple logical constraints. (See Appendix E for details.)

## 6. Conclusion

In this paper, we identify the pervasive unidirectional bias in Scene Graph Generation (SGG) and introduce the Mutual-Perspective Inverse Relations (MPIR) principle to formalize visual symmetry. To operationalize this principle, we propose the Reciprocal Perspective Calibration (RPC) framework for robust relation calibration alongside the Hypernym-Guided Prompt (HGP) method, which leverages LLM-distilled semantic hypernyms to abstract entity roles in textual prompts.

Although our current implementation primarily focuses on data-centric augmentation via the Adaptive Inverse-Relation Augmentation (AIRA) strategy, this paradigm opens up promising avenues for the community, such as constructing datasets with explicit inverse relationship annotations and establishing more rigorous benchmarking metrics for inverse consistency. Future work will focus on expanding RPC to semi-supervised and open-vocabulary settings, extending the MPIR principle from 2D to 3D spaces, and scaling from static images to dynamic videos. Designing explicit consistency loss functions around MPIR also stands as a viable path. Ultimately, we hope these efforts will

drive SGG to transition from unidirectional statistical fitting toward bidirectional logical reasoning.

## Acknowledgements

This work is supported by the National Natural Science Foundation of China (62576006, 62076005, 62472004, 61860206004), Anhui Provincial University Natural Science Research Major Project (2025AHGXZK20027), and Provincial Quality Project of Education in the New Era in 2023 (Postgraduate Education 2023lhpysfjd009). We sincerely thank Professor Bin Luo for his generous support. Also, the authors acknowledge the High-performance Computing Platform of Anhui University for providing computing resources.

## Impact Statement

This research presents work whose goal is to advance the field of Machine Learning. There are many potential societal consequences of our work, none which we feel must be specifically highlighted here.

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

# A. Additional Implementation and Experimental Details

This appendix provides supplementary details regarding the optimization framework, implementation specifics, baseline configurations, and dataset categorization to facilitate the reproducibility of our results.

## A.1. Optimization Framework

To enable perspective-invariant learning, we optimize the model over the augmented dataset $\mathcal{D}_{rpc}$, which encompasses both primal and reciprocal views.

**Objective Function.** The training objective is to minimize the expected negative log-likelihood (Cross-Entropy loss):

$$\mathcal{L} = \mathbb{E}_{\langle s, p_{gt}, o \rangle \sim \mathcal{D}_{rpc}} \left[ -\log P(p_{gt}|s, o) \right], \tag{8}$$

where $p_{gt}$ represents the ground-truth predicate label. The calculation of the probability $P(p_{gt}|s, o)$ depends on the underlying model architecture.

**Probability Calculation.** We instantiate the probability computation mechanism tailored to two distinct architectural paradigms:

- **Closed-Set Models (Linear Head):** For architectures utilizing a fixed vocabulary $\mathcal{C}$ (e.g., Motifs, VCTree), the probability is derived from a linear classifier. Let $\mathbf{f}_v \in \mathbb{R}^d$ denote the context-aware visual feature extracted by the backbone. The probability for class $c$ is computed as:

$$P(c|s, o) = \text{softmax}(\mathbf{W}\mathbf{f}_v + \mathbf{b})_c = \frac{\exp(\mathbf{w}_c^\top \mathbf{f}_v + b_c)}{\sum_{j \in \mathcal{C}} \exp(\mathbf{w}_j^\top \mathbf{f}_v + b_j)}, \tag{9}$$

where $\mathbf{W} \in \mathbb{R}^{|\mathcal{C}| \times d}$ and $\mathbf{b} \in \mathbb{R}^{|\mathcal{C}|}$ are the learnable weight matrix and bias vector.

- **Open-Set/CLIP Models (Similarity Head):** For CLIP-enhanced architectures utilizing HGP, prediction relies on vision-language alignment. The probability is predicted via temperature-scaled cosine similarity between $\mathbf{f}_v$ and the text embedding $\mathbf{z}_c$ derived from the prompt:

$$P(c|s, o) = \frac{\exp(\text{sim}(\mathbf{f}_v, \mathbf{z}_c)/t)}{\sum_{j \in \mathcal{C}} \exp(\text{sim}(\mathbf{f}_v, \mathbf{z}_j)/t)}, \tag{10}$$

where $t$ is a learnable temperature parameter scaling the logits.

By optimizing Eq. 8 using either Eq. 9 or Eq. 10, our RPC framework effectively enforces the logical consistency of inverse relations regardless of the backbone architecture.

## A.2. Implementation Specifics and Hyperparameters

**Backbone and Optimization.** All experiments are conducted using the CLIP framework with a ViT-B/32 visual backbone. Following the protocols established by HP (Zhu et al., 2024), the visual encoder remains frozen throughout the training process to preserve the robust generalization capabilities of the pre-trained representations. Trainable parameters are strictly limited to the learnable context vectors and the lightweight adaptation layers integrated into the text encoder, ensuring parameter efficiency and numerical stability.

**Hyperparameter Configurations.** Regarding specific hyperparameter configurations, we set the stochastic swapping probability $\lambda$ to 0.5 across all main experiments to ensure balanced exposure to primal and inverse perspectives. For the soft prompting component (HGP), the number of learnable context vectors, denoted as $ctx\_n$, is fixed at 2. While our framework supports flexible prompt insertion positions (including prefix, midfix, and suffix), we exclusively utilize the prefix configuration for all reported results unless otherwise specified.

### A.3. Baseline Architectures and Comparative Paradigms

**Baseline Models.** Our framework is integrated into several representative architectures, including **PE-Net** (Zheng et al., 2023), a strong baseline that leverages prototype-based embeddings for semantic regularization to enhance predicate classification.

**CLIP-Enhanced Models.** We introduce $CLIP_M$ and $CLIP_V$, which are variants derived from the Motifs and VCTree architectures, respectively. In these models, the original relation classification heads are replaced by the CLIP model, enabling them to leverage VLM priors for relationship reasoning.

**Competitive Paradigms.** We evaluate our RPC framework against two primary categories of established SGG paradigms:

- **Unbiased Learning Strategies:** Including causal inference and re-balancing approaches such as TDE (Tang et al., 2020), IETrans (Zhang et al., 2022), ST-SGG (Kim et al., 2024b), and the transformer-based VETO (Sudhakaran et al., 2023). These methods aim to mitigate the severe long-tail distribution in SGG, where models tend to over-predict high-frequency predicates (e.g., *on*, *near*) while neglecting semantically rich tail classes. Specifically, **TDE** leverages causal inference to remove contextual bias, **IETrans** addresses internal label bias via sample transfer, **ST-SGG** introduces adaptive spatio-temporal constraints, and **VETO** exploits global attention to model complex predicate dependencies.

- **Knowledge-Driven Models:** Which utilize external priors, including GSGG (Zhu et al., 2025), HP (Zhu et al., 2024), MLLMP (Songju et al., 2025), and CooK (Kim et al., 2024a). These frameworks incorporate external knowledge to enhance semantic understanding, especially when visual features alone are insufficient for disambiguating complex relations. For instance, **GSGG** builds a generic architecture with common-sense constraints, **HP** introduces hierarchical soft prompting to refine predicate learning, **MLLMP** utilizes multimodal large language models as knowledge engines, and **CooK** integrates global co-occurrence knowledge with learnable TF-IDF priors to boost performance.

### A.4. Predicate Categorization

The specific classification of predicates based on their frequency of occurrence (i.e., long-tailed distribution) is presented in Table 6. This categorization strictly follows the experimental code provided by HP (Tang, 2019; Zhu, 2026).

*Table 6.* Predicate categories used in VG. Predicates are grouped into head, body, and tail according to the long-tailed distribution.

| Category | Predicate |
|---|---|
| head | holding, riding, sitting on, standing on, has, in, with, behind, wearing, near, in front of, of, wears, above, under, on |
| body | attached to, carrying, covered in, covering, hanging from, laying on, looking at, parked on, using, walking on, watching, for, belonging to, at, over, between, and |
| tail | eating, flying in, growing on, lying on, mounted on, painted on, playing, says, walking in, from, to, made of, against, part of, across, on back of, along |

It is important to note that the categorization based on long-tailed distribution presented here is derived exclusively from the empirical data statistics. Consequently, this process functions independently of LLM. In contrast, the semantic categorization and mapping functions, which leverage LLM as auxiliary tools to resolve ambiguity, will be detailed in the subsequent sections.

## B. Handling Closed-Set Constraints and Architecture Adaptations

In this section, we elaborate on the implementation strategies for integrating the RPC framework into closed-set SGG baselines, including Motifs, VCTree, and PeNet. Unlike open-set architectures such as CLIP, which possess the inherent capability to encode arbitrary semantic concepts, standard closed-set baselines operate under the constraint of a predefined predicate vocabulary $\mathcal{C}$. To accommodate this fundamental limitation, we propose two complementary strategies. The **Filtering Strategy**, employed in our primary closed-set experiments, ensures vocabulary compliance by selectively retaining only in-vocabulary inverse predicates. Alternatively, the **Expansion Strategy** offers a mechanism for handling out-of-distribution (OOD) predicates through controlled vocabulary augmentation.

## B.1. Filtering Strategy: Vocabulary-Constrained AIRA

The primary challenge in applying AIRA to closed-set models lies in the generation of out-of-distribution (OOD) inverse predicates. For a generated inverse triplet $\tau^{-1} = \langle o, p^{-1}, s \rangle$, if the inverse predicate satisfies $p^{-1} \notin \mathcal{C}$, compelling the model to predict such an unseen predicate would inevitably result in classification failure.

To mitigate this issue, we employ a **Validity Filter** that enforces vocabulary compliance. Specifically, a generated inverse triplet is retained if and only if its inverse predicate resides within the predefined vocabulary. Formally, the augmented training set is constructed as

$$\mathcal{D}_{rpc} = \{\langle o, p^{-1}, s \rangle \mid \langle s, p, o \rangle \in \mathcal{D}_{train} \wedge p^{-1} \in \mathcal{P}\}. \tag{11}$$

**Unified Augmentation Protocol.** In the open-set setting (Sec. 4.1), we impose a strict distinction between Type I (Head) and Type II (Tail) relations to deploy tailored strategies (Additive Injection vs. Stochastic Replacement). However, in the closed-set regime, the Validity Filter substantially reduces the cardinality of admissible inverse candidates. Enforcing a rigid type-based dichotomy under such constraints may yield insufficient reciprocal supervision signals, thereby undermining the effectiveness of inverse augmentation.

To address this sparsity issue, we relax the type-specific constraints for closed-set experiments and adopt a **Unified Injection Strategy**. Rather than conditioning strategy selection on head/tail distributional characteristics, we apply AIRA uniformly to **all** valid in-vocabulary inverse pairs. This approach ensures maximal exposure to reciprocal relational logic within the confines of the predefined vocabulary, effectively compensating for the sample scarcity induced by OOD filtering.

## B.2. Expansion Strategy: Model-Specific Architectural Adaptations

We delineate the architectural modifications necessary to enable closed-set baselines to accommodate vocabulary expansion under the Expansion Strategy. For each model, we first characterize the structural constraints inherent to its original design, then detail the corresponding adaptation protocols required to support the expanded vocabulary $\mathcal{C}' = \mathcal{C} \cup \mathcal{P}_{\text{inverse}}$.

### 1. Motifs (Neural Motifs)

**Original Constraint: Fixed Frequency Prior and Static Classification Head.** The Neural Motifs architecture fundamentally relies on a statistical frequency prior, instantiated as a fixed tensor $B \in \mathbb{R}^{|\mathcal{C}| \times |\mathcal{C}| \times |\mathcal{P}|}$. This tensor encodes the empirical conditional distribution $P(p \mid s, o)$ derived from training set co-occurrence statistics. The final relationship classifier comprises a fully connected (FC) layer with output dimensionality fixed at $|\mathcal{P}|$, corresponding to the original vocabulary. Consequently, the architecture inherently rejects out-of-vocabulary predictions, as novel predicates possess undefined (zero-valued) frequency entries and lack corresponding classifier output nodes.

**Adaptation Protocol: Frequency Bias Augmentation and Classifier Expansion.** To enable prediction of novel inverse relations, we implement a two-stage adaptation procedure.

- **Frequency Bias Tensor Expansion.** We expand the bias tensor $B$ along its predicate dimension to shape $|\mathcal{C}| \times |\mathcal{C}| \times |\mathcal{P}'|$. Since novel inverse predicates (e.g., *eaten by*) are absent from the original training distribution, their empirical frequencies cannot be directly computed. Rather than initializing these entries to zero—which would impose prohibitive penalties on their prediction—we initialize the bias values for novel predicates to the mean frequency of tail-class predicates. This neutral initialization mitigates the suppressive effect of the statistical prior on inverse relation learning.

- **Classifier Head Resizing.** We reinitialize the final FC layer parameters from $W \in \mathbb{R}^{d \times |\mathcal{P}|}$, $b \in \mathbb{R}^{|\mathcal{P}|}$ to $W' \in \mathbb{R}^{d \times |\mathcal{P}'|}$, $b' \in \mathbb{R}^{|\mathcal{P}'|}$. Parameters corresponding to original predicates are retained from any available pretraining, while newly introduced parameters are initialized via Xavier uniform initialization.

### 2. VCTree (Visual Context Tree)

**Original Constraint: Dynamic Hierarchical Encoding with Fixed-Dimension Scoring Head.** VCTree employs a TreeLSTM architecture to encode hierarchical visual contexts through dynamic tree construction. While the tree topology generation is intrinsically class-agnostic and depends solely on visual features, the subsequent scoring module—comprising a bidirectional TreeLSTM decoder coupled with a fixed-dimension classification head—imposes vocabulary constraints. The architectural rigidity resides in this final decoding-to-classification pipeline, which is dimensionally bound to the original predicate space $|\mathcal{P}|$.

**Adaptation Protocol: Topology-Invariant Classification Head Expansion.** Our adaptation preserves the hierarchical context modeling mechanism while extending predictive capacity.

- **Preservation of Tree Construction Logic.** The tree topology generation module, including the connection validity scoring function (binary edge classification) and the structure optimization procedure, remains unmodified. This ensures consistency with the original hierarchical visual reasoning framework.

- **Classification Head Expansion.** We expand the output dimensionality of the final classifier layer, which operates on TreeLSTM decoder hidden states, from $|\mathcal{P}|$ to $|\mathcal{P}'|$. Critically, unlike Neural Motifs, VCTree does not employ hard-coded frequency statistics during inference. The adaptation is therefore purely architectural, involving only the resizing of the terminal linear projection layer. This enables the learned hierarchical context representations to directly supervise the acquisition of novel inverse semantics through gradient-based optimization, without interference from distributional priors.

## 3. PE-Net (Prototype-based Embedding Network)

**Original Constraint: Fixed-Cardinality Prototype Bank.** PE-Net diverges from conventional discriminative classifiers by employing a prototype-based metric learning paradigm. It maintains a learnable prototype matrix $M \in \mathbb{R}^{|\mathcal{P}| \times D}$, where each row vector encodes the embedding representation of a distinct predicate class. Classification is performed via similarity measurement (e.g., cosine distance) between input visual features and these prototype vectors. The architectural constraint is imposed by the fixed cardinality of the prototype bank, which is dimensionally restricted to $|\mathcal{P}|$.

**Adaptation Protocol: Semantically-Initialized Prototype Augmentation.** PE-Net presents a distinctive advantage for vocabulary expansion due to its embedding-based formulation. We implement the following adaptation procedure.

- **Prototype Bank Expansion with Semantic Initialization.** We extend the prototype matrix from $M \in \mathbb{R}^{|\mathcal{P}| \times D}$ to $M' \in \mathbb{R}^{|\mathcal{P}'| \times D}$. Rather than employing random initialization, we exploit the inherent semantic structure of inverse relations. For each novel inverse predicate $p^{-1}$, we derive its semantic embedding via a pretrained language model (e.g., GloVe or BERT) and utilize this representation to initialize the corresponding prototype vector in $M'$. Existing prototype vectors for original predicates are retained from pretraining when available.

- **Advantages of Semantic Initialization.** This semantically-grounded initialization strategy provides a principled warm-start for expanded predicate classes. Even when visual training instances for an inverse relation are scarce (characteristic of long-tail distributions), the model can leverage the semantic alignment between prototype embeddings and visual-semantic contexts to facilitate accelerated convergence. This contrasts favorably with the random initialization required for architectures such as Motifs or VCTree, which lack explicit semantic grounding in their classification heads.

**Summary.** We emphasize that both proposed strategies rigorously uphold the model-agnostic design philosophy underlying the RPC framework, as detailed below.

*Adherence to Model-Agnostic Design Principles.* The **Filtering Strategy** adheres to the strictest interpretation of a plug-and-play module, requiring zero architectural modifications to the host model. The **Expansion Strategy**, while necessitating model-specific adaptations, restricts these modifications to **minimal interface reconfigurations**—such as resizing the final classification head or expanding the prototype bank—rather than fundamental architectural redesigns. Critically, the core reasoning mechanisms of each architecture remain structurally preserved, including the message-passing dynamics in Motifs, the TreeLSTM hierarchical encoding in VCTree, and the prototype-based metric learning in PE-Net. Consequently, both strategies maintain the essential properties of **generality, minimal intrusiveness, and architectural independence** requisite for a truly model-agnostic framework.

*Implications for Open-Vocabulary Generalization.* The RPC framework exhibits theoretical universality, rendering it applicable to arbitrary open-set architectures beyond the specific baselines evaluated herein. Moreover, the proposed *Inverse Consistency Metrics* (e.g., IR@K, MI@K) function not merely as measures of logical symmetry, but also as diagnostic indicators of **open-vocabulary competence**. Since logical inversions frequently involve predicates that are semantically distinct from the original predicate set—often manifesting as long-tail or out-of-distribution relations—a model's performance on inverse consistency metrics inherently reflects its capacity to generalize to novel semantic concepts under open-world conditions. Thus, strong inverse consistency serves as both a validation of relational reasoning integrity and a proxy for robust semantic generalization.

## C. Implementation Details of LLM-Driven Components

The external semantic knowledge base is constructed using Gemini-2.5. This knowledge base serves three primary purposes: (1) categorizing each predicate into Type I, II, or III for tailored augmentation; (2) generating logically consistent inverse predicates for the MPIR setting (Mapping Function $\Phi$); and (3) mapping the 150 fine-grained object categories in Visual Genome into $J = 15$ coarse-grained semantic hypernyms for HGP (Mapping Function $\Psi$). Below, we detail the implementation of each component.

### C.1. LLM-Driven Taxonomy Refinement

While our predicate taxonomy aligns with established definitions in Neural Motifs (Zellers et al., 2018) and recent advances (Jiang et al., 2025), we do not strictly adhere to their static partitions. Instead, we employ an **LLM-driven refinement process** to adapt these categories for reciprocal calibration.

To mitigate the stochastic hallucinations inherent in Large Language Models (LLMs) and ensure reproducibility, we implement a **reference-guided prompting strategy**. Rather than soliciting unconstrained clustering, we provide the classification sets from prior works (Zellers et al., 2018; Jiang et al., 2025) as *initial anchor values* within the context. This setup serves as a reference prior, guiding the LLM to refine the boundaries and generate optimized super-categories ($\mathcal{SC} = \{\text{SEMANTIC}, \text{GEOMETRIC}, \text{POSSESSIVE}\}$) that maximize the efficacy of our AIRA strategy. The specific prompt template employed is presented in Fig. 4.

---

**Prompt Template for Predicate Categorization**

**Role:** You are an expert in Scene Graph Generation and linguistic semantics.
**Task:** Classify the given list of 50 visual predicates into three specific super-categories: SEMANTIC (Action/Interaction), GEOMETRIC (Spatial), and POSSESSIVE (Part-whole/Ownership).

**Reference Anchors:**

- SEMANTIC: [eating, riding, holding, walking on, hanging from, mounted on, ...]

- GEOMETRIC: [on, under, behind, near, in front of, on back of, ...]

- POSSESSIVE: [has, of, belonging to, made of, part of, ...]

**Constraints:** 1. **Geometric:** Strictly for spatial relationships reversible by logic (e.g., above/below). 2. **Possessive:** Strictly for ownership or body parts (e.g., has, part of). 3. **Semantic:** For all other dynamic interactions. 4. Use the provided references as a baseline but correct any semantic inconsistencies suitable for inverse mapping.
**Input Predicates:** [List of 50 predicates...]
**Output Format:** JSON list with "predicate": "category".

---

*Figure 4.* The structured prompt template used to guide the LLM in refining predicate categories, utilizing prior works as initialization anchors to ensure stability.

### C.2. Instantiation of Mapping Functions

While the MPIR framework provides a theoretically complete foundation for understanding relational reciprocity, its practical implementation necessitates precise definitions for the abstract mapping functions. As posited in the main text, a relationship represents a unique, "bi-directionally locked" physical fact. However, a predicate is merely a directional linguistic description of this fact, anchored to a specific subject viewpoint. Consequently, the inverse predicate corresponding to a given primal description is not always uniquely determined by simple heuristic rules. For instance, the visual fact of a person holding a phone could linguistically map to $\langle \text{phone}, \text{held by}, \text{person} \rangle$ or $\langle \text{phone}, \text{in hand of}, \text{person} \rangle$.

To resolve this semantic ambiguity and ensure rigorous experimentation, the inverse relation mapping function $\Phi$ and the semantic hypernym projection function $\Psi$ must be concretely instantiated. Instead of relying on rigid, handmade rules which may lack semantic nuance, we leverage the rich, open-world semantic priors inherent in LLMs to distill these functions. This process involves an **empirical calibration** step, where LLM outputs are verified against visual consistency criteria, ensuring an optimal balance between linguistic precision and token efficiency.

C.2.1. INVERSE RELATION MAPPING FUNCTION ($\Phi$)

The primary challenge in defining $\Phi : p \to p^{-1}$ lies in handling the potential non-uniqueness of linguistic inverses. To establish a deterministic mapping for our experimental protocol, we prompted the LLM to identify the most distinct and visually representative inverse counterpart for each predicate within the Visual Genome vocabulary. The prompting strategy emphasized that the inverse predicate must describe the identical physical interaction but strictly from the perspective of the original object acting as the new subject.

Based on the intrinsic reciprocity of physical interactions, we categorize the mapping logic into two primary regimes. For **Symmetric** relations, the inverse predicate remains identical to the primal one (Identity Mapping). For **Asymmetric** relations, we further distinguish three specific subtypes to apply precise inversion rules. Table 7 provides illustrative examples of the instantiated $\Phi$ mapping.

*Table 7.* Examples of the instantiated Inverse Mapping Function $\Phi : p \to p^{-1}$. The mappings are categorized into Symmetric (Identity) and Asymmetric (Geometric, Possessive, Semantic) types.

| Geometric & Possessive | | | Semantic & Symmetric | | |
|---|---|---|---|---|---|
| Type | Primal ($p$) | Inverse ($p^{-1}$) | Type | Primal ($p$) | Inverse ($p^{-1}$) |
| Geometric | `on` | `under` | Semantic | `wearing` | `worn by` |
| Geometric | `behind` | `in front of` | Semantic | `eating` | `eaten by` |
| Geometric | `above` | `below` | Semantic | `covering` | `covered in` |
| Possessive | `has` | `part of` | Symmetric | `near` | `near` |
| Possessive | `belonging to` | `owns` | Symmetric | `and` | `and` |

As illustrated in Fig. 5, we formulated the task as a linguistic transformation problem guided by a specific taxonomy. We explicitly categorized the visual relationships into four distinct types to address the diversity of the Visual Genome vocabulary. For each category, we provided two curated **few-shot examples** to steer the LLM's reasoning.

---

**Prompt Template for Inverse Relation Mapping ($\Phi$)**

**Role:** You are an expert linguist specializing in Visual Relationship Detection (VRD).
**Task:** Generate the inverse predicate ($p^{-1}$) for a given primal predicate ($p$). Ensure the inverted triplet $\langle \text{Object}, p^{-1}, \text{Subject} \rangle$ describes the **identical physical fact** as the primal $\langle \text{Subject}, p, \text{Object} \rangle$.
**Taxonomy & Few-Shot Examples:**

- SYMMETRIC (Identity Mapping):    Input: `and` → Output: `and`;    Input: `with` → Output: `with`

- SEMANTIC (Action / Passive):    Input: `using` → Output: `used by`;    Input: `covered in` → Output: `covering`

- GEOMETRIC (Spatial Opposition):    Input: `on` → Output: `under`;    Input: `above` → Output: `below`

- POSSESSIVE (Ownership Inversion):    Input: `part of` → Output: `has`;    Input: `belonging to` → Output: `own`

**Input Predicates:** [List of target predicates...]
**Output Format:** JSON format with "primal": "inverse".

---

*Figure 5.* **Structured Prompt for $\Phi$ Instantiation.** We employ an In-Context Learning approach with an LLM, utilizing a four-category taxonomy to guide the generation of visually consistent inverse predicates.

This structured few-shot approach effectively resolves ambiguities. For instance, without the "Geometric" context, an LLM might map *on* to *off* (antonym) rather than *under* (spatial inverse). By anchoring the generation with these category-specific prototypes, we obtained a high-quality mapping for all 50 predicates in the VG dataset, which was subsequently manually verified for visual consistency.

C.2.2. SEMANTIC HYPERNYM PROJECTION FUNCTION ($\Psi$)

The hypernym projection function $\Psi : \mathcal{V} \to \mathcal{H}$ is designed to categorize the fine-grained entity vocabulary $\mathcal{V}$ into a compact set of hypernyms $\mathcal{H}$, essential for the Hypernym-Guided Prompt (HGP) to prevent combinatorial explosion. Similar to $\Phi$, we utilized the LLM to analyze the semantic attributes of all 150 object categories in the VG dataset.

The LLM was instructed to perform a functional abstraction: mapping entities not just by biological taxonomy, but by their role in visual interactions. For example, while `boy` and `girl` are biologically distinct, they share the identical functional role of "human agent" in relationships like `riding` or `holding`. Therefore, they are projected to the same hypernym. This abstraction reduces the search space while preserving the critical subject-object context required for relation detection. Furthermore, empirical observations reveal that the initial object detection stage frequently suffers from label inconsistency and redundant bounding box generation. For instance, a single visual entity might be redundantly labeled as both "man" and "person" within the same scene, or semantically identical objects (e.g., `surfboards`) might be arbitrarily assigned different granularities, such as one being labeled `board` and another `surfboard`. By compressing these fine-grained details into unified functional categories, our semantic hypernym projection effectively mitigates **such detector noise**, ensuring that the relational reasoning focuses on the core semantic roles rather than being distracted by label synonyms or inconsistent granularities. We define a total of $J = 15$ coarse-grained hypernyms. Representative mappings are shown in Table 8.

*Table 8.* Examples of the Semantic Hypernym Projection $\Psi : \mathcal{V} \to \mathcal{H}$. Fine-grained entities are mapped to coarse-grained functional hypernyms.

| Hypernym ($h \in \mathcal{H}$) | Mapped Entities ($v \in \mathcal{V}$) |
| --- | --- |
| person | man, woman, boy, girl, child, guy, lady, player... |
| animal | dog, cat, horse, sheep, elephant, zebra, bird... |
| vehicle | car, bus, train, truck, bike, motorcycle, airplane... |
| furniture | chair, table, bed, bench, desk, shelf... |
| clothing | shirt, pants, jacket, hat, shoe, shorts, jean, cap, sock, tie... |

---

**Prompt Template for Hypernym Projection ($\Psi$)**

**Role:** You are an expert taxonomist specializing in Visual Scene Graph generation.
**Task:** Map 150 fine-grained object categories ($\mathcal{V}$) into 15 functional hypernyms ($\mathcal{H}$). Perform functional abstraction by grouping entities based on their interaction roles rather than strict biology (e.g., `boy`, `girl` → PERSON) and normalizing specific tools into functional super-categories (e.g., `surfboard` → VEHICLE).
**Target Taxonomy & Few-Shot Anchors:**

- PERSON: Input: `boy`, `girl`, `man`, `skier`, `child` → Output: `Person`

- ANIMAL: Input: `horse`, `dog`, `sheep`, `cow`, `bear` → Output: `Animal`

- VEHICLE: Input: `car`, `bus`, `surfboard`, `bike`, `boat` → Output: `Vehicle`

- FURNITURE: Input: `table`, `chair`, `bed`, `desk`, `cabinet` → Output: `Furniture`

- CLOTHING: Input: `shirt`, `pants`, `hat`, `jacket`, `shoe` → Output: `Clothing`

**Input Objects:** [List of 150 VG objects...]
**Output Format:** JSON format with "object": "hypernym".

---

*Figure 6.* **Structured Prompt for $\Psi$ Instantiation.** We instruct the LLM to perform functional abstraction, mapping fine-grained entities to 15 coarse-grained hypernyms to prevent combinatorial explosion in prompt construction.

Crucially, the tasks facilitated by the LLM were not conducted in isolation as a purely automated process. During the generation of results and the resolution of semantic inquiries, we actively corroborated the model's outputs with insights from domain experts and experienced practitioners. This rigorous cross-verification process serves to fully validate the effectiveness and credibility of the LLM, ensuring that its contributions are both logically sound and domain-accurate. Serving as a robust auxiliary tool, the LLM significantly streamlines the experimental workflow, enhancing both efficiency and comprehensiveness. While we acknowledge the potential for hallucinations, rigorous human oversight and critical verification can effectively mitigate these risks. We encourage future researchers building upon this work to leverage LLM to further refine and expand these methodologies.

# D. Algorithm

---

**Algorithm 1** RPC Framework: AIRA and HGP Training

---

**Require:** Training set $\mathcal{D}_{train}$, Predicate set $\mathcal{P}$, Visual Encoder $f_\theta$.
**Require: Knowledge Base:** Inverse map $\Phi : p \rightarrow p^{-1}$, Hypernym map $\Psi : \mathcal{V} \rightarrow \mathcal{H}$.
**Ensure:** Optimized parameters $\theta^*$ and learnable context $\mathbf{U}$.
   Initialize calibrated dataset $\mathcal{D}_{rpc}$   *// Phase 1: AIRA* $\leftarrow \emptyset$.
  **for** each triplet $\tau = \langle s, p, o \rangle \in \mathcal{D}_{train}$ **do**
     Generate inverse triplet $\tau^{-1} = \langle o, \Phi(p), s \rangle$.
     **if** $p \in$ Type I $\cup$ Tail **then**
        *Additive Injection:* $\mathcal{D}_{rpc} \leftarrow \mathcal{D}_{rpc} \cup \{\tau, \tau^{-1}\}$.
     **else if** $p \in$ Type II $\cup$ Head **then**
        *Stochastic Replacement:* Sample $\delta \sim$ Bernoulli$(\lambda)$.
        $\mathcal{D}_{rpc} \leftarrow \mathcal{D}_{rpc} \cup \{\delta \cdot \tau^{-1} + (1 - \delta) \cdot \tau\}$.
     **else**
        $\mathcal{D}_{rpc} \leftarrow \mathcal{D}_{rpc} \cup \{\tau\}$.
     **end if**
  **end for**
  *// Phase 2: Hypernym-Guided Prompt Optimization*
  **repeat**
     Sample mini-batch $\mathcal{B} \sim \mathcal{D}_{rpc}$.
     **for** each $\langle s, p, o \rangle \in \mathcal{B}$ **do**
        Get hypernyms $h_s = \Psi(s), h_o = \Psi(o)$.
        Construct HGP $\mathcal{T} = [\mathbf{U}, \boldsymbol{e}_{h_s}, \boldsymbol{e}_p, \boldsymbol{e}_{h_o}]$.
        Compute probability $P(p|s, o)$ via Eq. (10).
     **end for**
     $\mathcal{L} = \sum_{\mathcal{B}} -\log P(p_{gt}|s, o)$.
     Update $[\theta, \mathbf{U}] \leftarrow$ Optimizer$(\nabla \mathcal{L})$.
  **until** convergence
  **return** $\theta^*, \mathbf{U}$

---

# E. Additional Ablation Studies

The AIRA strategy effectively mitigates the potential performance loss of RPC on the primal unidirectional manifold evaluation by disentangling the augmentation protocol. Specifically, for relations with high information density or low frequency, AIRA adopts the Add strategy; conversely, for those with low information density or high frequency, it employs the Rep strategy. By explicitly balancing information richness and frequency bias, RPC enhances the model's generalized understanding of visual interactions, resulting in performance improvements on both the primal and inverse manifolds. Note that the Possessive and Body categories not mentioned in the text are not ignored; rather, they are excluded because they demonstrated suboptimal performance in ablation studies.

To comprehensively validate the generalization of our method, we extended our ablation studies to two distinct architectures representing different paradigms: **PE-Net** (a representative closed-set model) and **CLIP$_\mathbf{M}$** (an open-set baseline). We specifically investigated the impact of different augmentation strategies (Add vs. Rep) on these architectures.

### E.1. Ablation on Closed-Set Baseline (PE-Net)

As shown in Table 9, we observe nuanced differences in the behavioral patterns of the closed-set PE-Net compared to open-set models. Consistent with our main findings, the proposed **AIRA** strategy (applying Add to Tail/Semantic and Rep to Head/Geometric) still yields the best overall performance, balancing R@K and mR@K effectively. However, applying the Add strategy globally to all categories (All) also achieves competitive results on PE-Net. We hypothesize that this phenomenon stems from the intrinsic nature of the closed-set setting. In closed-set training, the predicate vocabulary is fixed, and potential OOD predicates are naturally filtered out or mapped to background classes. Consequently, the Add operation—even when applied to Head/Geometric relations—functions as a fundamental data augmentation technique.

*Table 9.* Side-by-side ablation study comparing augmentation strategies on the closed-set PE-Net baseline. The left panel analyzes the **Add** strategy, while the right panel analyzes the **Rep** strategy. Baseline performance is repeated on both sides for direct comparison.

| Add | R@K | | mR@K | | M@K | | Rep | R@K | | mR@K | | M@K | |
|---|---|---|---|---|---|---|---|---|---|---|---|---|---|
| | 50 | 100 | 50 | 100 | 50 | 100 | | 50 | 100 | 50 | 100 | 50 | 100 |
| PE-Net | 63.1 | 65.0 | 29.4 | 31.1 | 46.3 | 48.1 | PE-Net | 63.1 | 65.0 | 29.4 | 31.1 | 46.3 | 48.1 |
| + T | 56.7 | 59.1 | 39.2 | 41.5 | 47.9 | 50.3 | + H | 58.7 | 61.0 | 37.3 | 39.5 | 48.0 | 50.2 |
| + H | 53.1 | 55.1 | 42.6 | 44.2 | 47.8 | 49.6 | + T | 66.7 | 68.3 | 25.6 | 27.2 | 46.2 | 47.8 |
| + Se | 57.4 | 59.7 | 40.0 | 42.2 | 48.7 | 50.9 | + Ge | 59.6 | 62.0 | 35.6 | 38.3 | 47.6 | 50.1 |
| + Ge | 59.5 | 61.4 | 34.8 | 36.5 | 47.2 | 48.9 | + Se | 60.0 | 61.8 | 34.2 | 35.2 | 47.0 | 48.5 |
| + All | 54.7 | 57.0 | 42.4 | 44.7 | 48.5 | 50.9 | + All | 57.5 | 59.8 | 38.4 | 40.6 | 48.0 | 50.2 |
| + T&Se | 57.2 | 59.6 | 40.4 | 42.9 | 48.8 | 51.2 | + H&Ge | 57.4 | 59.7 | 40.0 | 42.2 | 48.7 | 51.0 |
| + T&B | 52.4 | 54.8 | 43.5 | 46.1 | 47.9 | 50.4 | + H&B | 51.9 | 54.4 | 42.8 | 45.8 | 47.4 | 50.0 |
| + Se&Po | 53.0 | 55.0 | 42.9 | 44.4 | 47.9 | 49.7 | + Ge&Po | 57.7 | 60.0 | 38.2 | 40.4 | 48.0 | 50.2 |

*Table 10.* Side-by-side ablation study comparing augmentation strategies on the open-set CLIP$_M$ baseline. The left panel analyzes the **Add** strategy, while the right panel analyzes the **Rep** strategy.

| Add | R@K | | mR@K | | M@K | | Rep | R@K | | mR@K | | M@K | |
|---|---|---|---|---|---|---|---|---|---|---|---|---|---|
| | 50 | 100 | 50 | 100 | 50 | 100 | | 50 | 100 | 50 | 100 | 50 | 100 |
| CLIP$_M$ | 64.5 | 67.5 | 15.3 | 17.4 | 39.9 | 42.5 | CLIP$_M$ | 64.5 | 67.5 | 15.3 | 17.4 | 39.9 | 42.5 |
| + T | 64.4 | 66.8 | 18.1 | 19.8 | 41.2 | 43.3 | + H | 66.7 | 68.1 | 18.1 | 19.1 | 42.4 | 43.6 |
| + Se | 65.5 | 67.1 | 19.6 | 21.0 | 42.6 | 44.0 | + Ge | 64.8 | 67.6 | 17.0 | 18.9 | 40.9 | 43.2 |
| + T & Se | 67.9 | 68.4 | 21.0 | 22.8 | 44.4 | 45.7 | + H & Ge | 68.6 | 69.1 | 19.3 | 21.3 | 43.9 | 45.2 |
| + All | 63.2 | 65.9 | 17.6 | 19.7 | 40.4 | 42.9 | + All | 64.2 | 67.3 | 16.3 | 18.3 | 40.3 | 42.8 |

## E.2. Ablation on Open-Set Baseline (CLIP$_M$)

In contrast to the closed-set scenario, the open-set CLIP$_M$ baseline (Table 10) reveals significantly higher sensitivity to augmentation strategy alignment. While the standard AIRA configuration achieves the optimal balance, specifically reaching **45.7%** M@100 for Add: T&Se and **45.2%** M@100 for Rep: H&Ge, misapplied strategies yield severe penalties.

Specifically, applying Add to high-frequency Geometric (Ge) categories exacerbates positional biases, leading to suboptimal gains. Conversely, using Rep on scarce Tail (T) categories discards valuable training signals, resulting in a noticeable degradation in R@100 (dropping from 67.5% to 66.8%). Notably, a naive "All-in" approach (Add: All or Rep: All) fails to surpass our targeted H/T/Ge/Se division, with M@100 performance plateauing below 43%. This empirical evidence confirms that in open-set environments, correctly aligning the augmentation mechanism (Add vs. Rep) with intrinsic predicate properties is paramount for effective relational knowledge transfer.

## F. Background and Our Work

### F.1. From Unidirectional Mapping to Reciprocal Interaction

Scene Graph Generation (SGG) was formally pioneered by Krishna et al. (2017), who defined the task as parsing visual scenes into directed graphs $\mathcal{G} = (\mathcal{V}, \mathcal{E})$ to bridge low-level perception and high-level reasoning. Early dominant approaches prioritized feature refinement through contextual message-passing mechanisms. Foundational works like IMP (Xu et al., 2017) and Graph R-CNN (Yang et al., 2018) introduced iterative propagation to update features along directed paths. Subsequent methods enriched this paradigm: Neural Motifs (Zellers et al., 2018) and GPS-Net (Lin et al., 2020) employed LSTMs and graph property sensing to encode global context; VCTree (Tang et al., 2019) constructed dynamic tree structures for hierarchical information flow; and BGNN (Li et al., 2021) utilized bipartite graph networks to balance head and tail predictions.

To capture more complex dependencies, KET (Chen et al., 2019) integrated external commonsense knowledge, while Transformer-based architectures like RelTR (Cong et al., 2021), SGTR (Li et al., 2022), and Stacked Hybrid (Dong et al., 2022) exploited global attention mechanisms to model long-range interactions directly from visual tokens. Most recently, PE-Net (Zheng et al., 2023) proposed prototype-based embeddings to regularize the semantic space, and CooK (Kim et al., 2024a) incorporated global co-occurrence knowledge with learnable TF-IDF priors to further boost performance. Recognizing the severe long-tail distribution in SGG datasets, other recent approaches have focused on unbiased training

strategies, including re-weighting (Lin et al., 2017; Cui et al., 2019) and causal inference debiasing (Tang et al., 2020).

**Critique of Unidirectional Bias.** Despite these architectural advancements, existing paradigms predominantly model relationships as static, subject-centric unidirectional mappings ($s \rightarrow o$). Whether through sequence-dependent encoding (Zellers et al., 2018; Tang et al., 2019) or attention mechanisms (Cong et al., 2021; Li et al., 2022), the feature refinement process is inherently anchored on the subject. This formulation neglects the **reciprocity** of physical interactions (i.e., the logical equivalence of $s \rightarrow o$ and $o \rightarrow s$), leading to a fundamental unidirectional bias where models fail to recognize relationships from the inverse perspective.

## F.2. Prompt Learning in SGG

The advent of CLIP (Radford et al., 2021) has revolutionized visual recognition by aligning images and text in a shared semantic space. Inspired by the success of Context Optimization (CoOp) (Zhou et al., 2022b) and CoCoOp (Zhou et al., 2022a) in image classification, researchers have begun adapting soft prompting to relation detection. In the realm of Open-Vocabulary SGG, pioneering works have explored various prompting strategies. He et al. (2022) utilized fixed prompt-based finetuning to handle unseen predicates; Yu et al. (2023) introduced visually-prompted language models to boost fine-grained recognition; and more recently, RAHP (Liu et al., 2025) and SDSGG (Chen et al., 2024) employed hierarchical or role-playing prompts to enhance semantic alignment. However, in the context of Standard SGG, HP (Zhu et al., 2024) stands as the first attempt to introduce learnable soft prompts (i.e., CoOp-style) to closed-set scenarios, designing a hierarchical structure to refine predicate learning.

Despite these advancements, a critical trade-off persists in prompt engineering for relations. As highlighted by HP (Zhu et al., 2024), concise templates (e.g., "a photo of [PRED]") suffer from semantic ambiguity, as predicate meanings are inherently context-dependent (e.g., *riding* implies distinct spatial configurations for different agents). Conversely, explicit enumeration of full triplets (e.g., "[SUBJ] [PRED] [OBJ]") resolves ambiguity but triggers a **combinatorial explosion** of the search space ($\mathcal{O}(|\mathcal{V}|^2)$), rendering inference computationally prohibitive. Our method (HGP) resolves this dilemma. Unlike previous approaches that rely on expensive full-triplet enumeration or learnable vectors, we leverage LLM to extract semantic hypernyms, injecting coarse-grained subject-object constraints into the prompt. This strategy effectively bridges the gap between semantic directionality and computational feasibility.

## F.3. Logical Consistency and Inverse Relations

KG structure real-world facts as directed graphs where logical connectivity is paramount. Embedding methods like RotatE (Sun et al., 2019), SimplE (Kazemi & Poole, 2018), and PairRE (Chao et al., 2021) explicitly model inverse relations ($r(x,y) \iff r^{-1}(y,x)$) through mechanisms such as geometric rotations or paired vectors. However, this logical consistency remains largely overlooked in the visual domain. Existing SGG models typically treat predicates as isolated labels, resulting in logically contradictory predictions upon viewpoint shifts. To bridge this gap, we are the first to introduce a strict **RPC** mechanism into SGG. By enforcing the MPIR principle, we enable a transition from mere statistical fitting toward learning robust, physically grounded visual facts.

## F.4. Conclusion and Future Work

In this paper, we introduced the **RPC** (Reciprocal Perspective Calibration) framework to mitigate the pervasive unidirectional bias in Scene Graph Generation (SGG). By enforcing the **MPIR** principle, our approach enables models—particularly those in open-set environments like CLIP—to achieve a more robust understanding of physical interactions through reciprocal reasoning. While our current implementation demonstrates potent utility even under constrained supervision, several promising avenues for future exploration remain.

Beyond our current predicate-specific **AIRA** strategy, there is significant potential for developing a more unified, adaptive augmentation mechanism. Future work could investigate the concurrent application of **Add** and **Rep** strategies or integrate a gating mechanism to dynamically optimize data injection based on real-time distribution. Furthermore, the RPC framework can be re-envisioned as a versatile tool for dataset optimization:

- **Semi-supervised Learning**: Leveraging inverse relations as high-fidelity pseudo-labels to double effective dataset scale at minimal cost.

- **Diagnostic Benchmarking**: Pre-annotating reciprocal pairs to establish rigorous standards for evaluating relational

logic.

- **Low-shot Generalization**: Utilizing reciprocal data as a powerful inductive bias for open-vocabulary and few-shot SGG tasks.

Ultimately, we contend that logically consistent scene graphs are essential for advancing downstream tasks such as visual reasoning and navigation. In conclusion, inverse relations represent a critical frontier for both enhancing and auditing the relational intelligence of future vision systems.

## G. Notation Summary

To facilitate a clearer understanding of the proposed framework, we summarize the key notations used throughout the paper in Table 11.

*Table 11.* Summary of mathematical notations used in HGP and RPC.

| Symbol | Description |
|---|---|
| *Problem Formulation & Logic* | |
| $\mathcal{G} = (\mathcal{V}, \mathcal{E})$ | The scene graph where $\mathcal{V}$ are object nodes and $\mathcal{E}$ are relationship edges. |
| $\tau = \langle s, p, o \rangle$ | A semantic triplet consisting of subject $s$, predicate $p$, and object $o$. |
| $\tau^{-1}$ | The logically equivalent inverse triplet $\langle o, p^{-1}, s \rangle$. |
| $p^{-1}$ | The inverse predicate corresponding to $p$ (e.g., *riding $\leftrightarrow$ ridden by*). |
| $\mathcal{R}$ | The physical visual reality or "visual fact" depicted by the image. |
| $\mathcal{D}_{train}$ | The original training dataset. |
| $\mathcal{D}_{rpc}$ | The reciprocal-calibrated dataset encompassing both primal and reciprocal views. |
| *RPC Framework (AIRA)* | |
| $\Phi(\cdot)$ | The bijective mapping function transforming a primal triplet to its inverse. |
| $\lambda$ | The probability threshold for the Stochastic Replacement strategy. |
| $\delta$ | Bernoulli random variable ($\delta \sim$ Bernoulli($\lambda$)) controlling view swapping. |
| $\tilde{\tau}$ | The final training triplet after applying AIRA. |
| *Hypernym-Guided Prompting (HGP)* | |
| $\mathcal{H}$ | The compact set of semantic hypernyms ($\{h_j\}_{j=1}^{J}$). |
| $\Psi(\cdot)$ | The projection function mapping fine-grained entities to hypernyms ($\mathcal{V} \rightarrow \mathcal{H}$). |
| $\mathbf{U}$ | The sequence of learnable context vectors $\{\boldsymbol{u}_1, \ldots, \boldsymbol{u}_M\}$. |
| $\boldsymbol{e}_{(\cdot)}$ | Fixed embedding vectors (for hypernyms or predicates) from the frozen CLIP encoder. |
| $\mathcal{T}_{org}, \mathcal{T}_{inv}$ | Constructed prompt templates for primal and inverse perspectives, respectively. |
| *Evaluation Metrics* | |
| IR@K | Inverse Recall at rank $K$, evaluating inverse relation recognition. |
| ImR@K | Inverse Mean Recall at rank $K$, averaged across all predicate categories. |
| IM@K | Inverse Mean at rank $K$, the composite metric calculated as $(\text{IR@K} + \text{ImR@K})/2$. |
| *Appendix Variables (Optimization)* | |
| $P(r\|s, o)$ | Predicted probability of relation $r$ given subject-object pair. |
| $f_v, z_r$ | Visual feature vector and text embedding vector for relation $r$. |
| $t$ | Temperature parameter scaling the cosine similarity in the loss function. |
| $\mathcal{L}$ | The negative log-likelihood (Cross-Entropy) loss objective. |
| $\text{sim}(\cdot)$ | Cosine similarity function. |

