# OpenReview forum: "Beyond Unidirectional Bias: Reciprocal Perspective Calibration in Scene Graph Generation"
_ICML.cc/2026/Conference — ICML 2026 regular_

### Official Review · Reviewer_wKzJ · 2026-03-10

**Soundness:** 3
**Presentation:** 3
**Significance:** 3
**Originality:** 3
**Overall Recommendation:** 5
**Confidence:** 4

**Summary:**

This paper investigates unidirectional bias in SGG, where relationships are conventionally treated as static, one-way mappings from subject to object that overlook the reciprocal nature of visual interactions. The authors introduce the Mutual-Perspective Inverse Relations principle and a model-agnostic Reciprocal Perspective Calibration framework. Within this framework, an Adaptive Inverse-Relation Augmentation strategy addresses long-tail distributions, while a Hypernym-Guided Prompt method balances semantic context with computational efficiency in Vision-Language Models. The study concludes with a new evaluation protocol for inverse consistency and experimental validation on the VG dataset.

**Compliance With Llm Reviewing Policy:**

Affirmed.

**Final Justification:**

The rebuttal effectively addresses my main concerns regarding the long-tail claims, hyperparameter tuning, and the handling of LLM hallucinations through proposed constraints. Therefore, I am inclined to accept this paper.

**Key Questions For Authors:**

My questions are below. Addressing these would help clarify the method's robustness and generalization capabilities.

1. To demonstrate the method's generalization beyond the current setup, could the authors provide results on the SGCls and ObjCls tasks, or evaluations on additional datasets such as OpenImages/PSG?
2. Given the reliance on Large Language Models, how does the framework handle potential hallucinations or ambiguities when defining inverse predicates? A discussion on the system's resilience to such upstream noise would be valuable.
3. What is the rationale behind the specific choices for the mixing ratio $\lambda$ and fixing the number of hypernyms $J$ at 15? Sensitivity analyses for these hyperparameters are needed to confirm that the gains stem from the method design rather than parameter tuning.
4. Regarding the long-tail performance, could you explain the observed decline in standard recall metrics (e.g., for PE-Net) and provide a fine-grained Head/Body/Tail breakdown to substantiate the claimed improvements on tail predicates?

**Limitations:**

yes

**Strengths And Weaknesses:**

**Strengths**

The submission addresses unidirectional bias in SGG through the Mutual-Perspective Inverse Relations (MPIR) principle and the Reciprocal Perspective Calibration (RPC) framework. The work demonstrates clear originality by formalizing the logical equivalence of inverse relations from knowledge graphs, providing a solid theoretical foundation for bidirectional reasoning. Practically, the RPC framework is designed as a model-agnostic module that integrates effectively with architectures like Motifs and VCTree. Furthermore, the manuscript is generally well-structured and articulates the theoretical motivations behind MPIR with clarity.

**Weaknesses**

However, the technical soundness and empirical validation present several weaknesses. The experimental scope is currently restricted to the PredCls protocol on the Visual Genome dataset, leaving the method's generalization capabilities on tasks like SGCls and ObjCls or datasets like OpenImages/PSG unclear. Additionally, the pipeline relies on Large Language Models to define inverse predicates and hypernyms, yet the manuscript does not analyze the potential impact of LLM ambiguities or hallucinations on system robustness.

The experimental design also lacks sensitivity analyses for key hyperparameters, specifically the mixing ratio $\lambda$ and the fixed number of hypernyms $J$, making it difficult to attribute performance gains solely to the proposed method. Finally, the claim regarding long-tail distribution mitigation is not fully supported by the empirical evidence. The integration of RPC coincides with a decline in standard recall metrics for certain models, such as the R@50 drop from 63.1 to 57.4 for PE-Net, and the text does not provide a fine-grained analysis (e.g., Head/Body/Tail breakdown) to substantiate the specific improvements on tail predicates.

---

> ### Author Rebuttal · Authors · 2026-03-31
>
> R1:Appreciate your feedback. As a debiasing SGG work, our core evaluation focuses on the PredCls task because it provides ground-truth bounding boxes and entity labels. This allows us to evaluate the predicate prediction capability in its purest, most isolated form. In contrast, SGCls and SGDet introduce compounding noise from object detectors, and ObjCls leans more toward object detection. We have previously conducted experiments on the SGCls and SGDet tasks. Their performance trends are generally similar to the fluctuations in PredCls, and the metric variations primarily stem from the performance improvements in PredCls. For the newly added results on the OI dataset, please refer to our response to Reviewer Hgck.
>
> R2:As detailed in Appendix C, we systematically explored various LLM strategies (e.g., zero-shot, MPIR-definition prompting, and few-shot with constraints) to generate inverse relations. To handle upstream noise and hallucinations, our framework deliberately employs prompt-constrained LLM generation:
>
> (1) Scalability vs. Manual Cost: While purely manual annotation is feasible, it is cost-prohibitive. We incorporate LLMs to ensure the scalability of our RPC framework, avoiding complete reliance on hand-crafted rules.
> (2) Unconstrained Oversimplification: Without constraints, LLMs fail to grasp inverse semantics and oversimplify results. For instance, they collapse the inverses of Semantic predicates like "riding on" → "under", or generate grammatical noise like "on" → "is on".
> (3) Ambiguity and Misclassification: Even when prompted with basic definitions, unconstrained LLMs struggle with non-uniqueness. For example, despite "above" being the canonical inverse of "below", the LLM stubbornly generates "on". This introduces ambiguity and artificially reduces the frequency of "above". Furthermore, without categorical restrictions, LLMs misclassify Possessive predicates as Semantic (e.g., generating "had by" for "have").
> Conclusion: To ensure system resilience against such upstream noise, we inject lightweight manual priors (introducing the MPIR concept and categorical constraints) into the prompts. This strictly regulates the LLM, effectively mitigating hallucinations, ambiguities, and fallacies.
>
> R3:The Rep strategy balances original and inverse relations via λ. Performance remains stable within a reasonable range, especially for open-set. For closed-set, λ is more sensitive: λ=0 degrades to the baseline, while λ=1 completely trades original metrics for inverse ones (though the overall average still slightly beats the baseline, as inverse relations inherently retain some original semantics). Note that the following table reports results for analysis setting λ on "$CLIP_{M} + RPC_{Rep}$"; F1 denotes the harmonic mean of R@100 and IR@100, and M denotes the arithmetic mean.
> | $\lambda$ | 0.00 | 0.25 | 0.50 | 0.75 | 1.00 |
> | :--- | :--- | :--- | :--- | :--- | :--- |
> | R@100 | 64.45 | 65.52 | 66.71 | 65.76 | 63.20 |
> | IR@100 | 51.67 | 53.09 | 53.92 | 54.41 | 55.56 |
> | F1 | 57.36 | 58.65 | 59.64 | 59.55 | 59.13 |
> | M | 58.06 | 59.31 | 60.32 | 60.09 | 59.38 |
>
> Regarding the sensitivity of the semantic hypernym count J:
> J is a structural setting with intrinsic semantic significance, not a trivially tunable hyperparameter. Altering J fundamentally re-partitions fine-grained classes; thus, preserving semantic integrity must outweigh arbitrary numerical adjustments. Nonetheless, we analyzed its impact:
> (1) Too small (J=7): Hypernyms become overly broad, diluting discriminative boundaries. The model confuses distinct predicates due to marginal subject-object semantic differences, dropping metrics like mR@K.
> (2) Too large (J=25): Semantic fragmentation occurs. Performance plateaus while the prompt space suffers a combinatorial explosion, sharply increasing GPU memory, latency, and training instability.
> This confirms our calibrated J is both necessary and effective.
>
> R4:In SGG, standard Recall (R@K) is instance-centric and heavily biased toward dataset head distributions. Therefore, Mean Recall (mR@K) is widely adopted to accurately measure long-tail performance. mR computes the average recall across all P predicate categories:
> $$\text{mR} = \frac{P_H}{P} \cdot \overline{R}_H + \frac{P_B}{P} \cdot \overline{R}_B + \frac{P_T}{P} \cdot \overline{R}_T$$
> where $P_i$ represents the number of predicate categories in each subset, and $\overline{R}_i$ is the average recall within that specific subset.
> Consequently, jointly evaluating R and mR is the standard indicator for tail improvements. To address your request and substantiate our claims, we provide a fine-grained Head/Body/Tail breakdown of mR@100 for PE-Net and our RPC framework (predicate splits detailed in Appendix A.4).
>
> | Method | Head | Body | Tail | mR |
> | :--- | :--- | :--- | :--- | :--- |
> | PE-Net | 43.78 | 44.41 | 23.38 | 37.06 |
> | + $RPC_{Rep}$ | 47.77 | 50.00 | 29.10 | 42.18 |
> | + $RPC_{Add}$ | 46.35 | 50.72 | 31.72 | 42.86 |
>
> Thanks for your concern.

---

> > ### Author Rebuttal · Reviewer_wKzJ · 2026-04-05
> >
> > All my core concerns have been completely resolved. I appreciate the authors' hard work during the rebuttal phase and will raise my score.

---

> > > ### Author Response · Authors · 2026-04-05
> > >
> > > We sincerely thank the reviewer for the highly positive comments and the generous decision to raise the score.
> > >
> > > Your strong endorsement is a tremendous encouragement to us, reinforcing our confidence that our proposed framework provides an effective and solid step forward for debiased generation. We sincerely hope that our proposed MPIR principle and our efforts to mitigate unidirectional bias can effectively address the persistent challenges in the SGG field. To make the work even more complete, we will ensure that all the supplementary details and clarifications discussed during this rebuttal phase are carefully incorporated into the appendix of the final version.
> > >
> > > Thank you once again for your invaluable guidance, time, and support in helping us improve this paper!

---

### Official Review · Reviewer_Hgck · 2026-03-12

**Soundness:** 3
**Presentation:** 3
**Significance:** 3
**Originality:** 3
**Overall Recommendation:** 4
**Confidence:** 3

**Summary:**

this paper proposes a new framework (RPC) to capture the bidirectional relations in scene graph generation. to fix the common issue where models only understand interactions in a single direction, the authors introduce a novel data augmentation strategy that trains the model on inverse relations. they also develop HGP to help VLMs process these dual perspectives efficiently. this approach ensures that models maintain logical consistency regardless of whether they are analyzing an interaction from the subject's or the object's point of view.

**Compliance With Llm Reviewing Policy:**

Affirmed.

**Final Justification:**

I appreciate the two-rounds explanation from the authors. My main concerns have been addressed. Thus, I'll keep my positive score.

**Key Questions For Authors:**

see weakenss

**Limitations:**

yes

**Strengths And Weaknesses:**

Strengths:

1. the authors focus on the unidirectional bias. standard models will learn "man riding horse" but completely fail to recognize the exact same image if you ask for the relation from the object's perspective ("horse ridden by man"). Forcing the model to learn the logical, physical inverse of relationships is a smart, physically grounded way to improve scene graphs.

2. RPC is a smart, logic-driven data augmentation strategy. since it operates mostly at the data and prompt level, it can work with existing API models or public models without redesigning the architecture. The HGP is also a clever method to keep prompt from exponentially exploding.

weaknesses:

1. the experiments rely entirely on the Visual Genome dataset. While VG is the standard for this task, it would make the paper much stronger to see if this reciprocal logic holds up in other, noisier real-world datasets.

2. a minor point about format. the running header on every page still says "Submission and Formatting Instructions for ICML 2026", where the ttile here is missing.

---

> ### Author Rebuttal · Authors · 2026-03-31
>
> **Q1: Evaluation on the Open Images (OI) Dataset**
>
> **R1:**
> We have incorporated the Open Images (OI) dataset to address this concern. The OI dataset is larger in scale, features more entities but fewer relationships, and contains greater annotation noise, making it a more challenging and realistic benchmark.
> First, unlike the dense annotations in VG, OI annotations are extremely sparse. Second, OI employs a more comprehensive set of evaluation metrics than VG. Beyond the standard R@K metrics, OI introduces an unannotated dictionary and places a stronger emphasis on precision. Specifically, the wmAP metric demands higher model confidence, and the overall score assigns greater weight to precision. The result is as follows.
> | Model | R@50 | mR@50 | M@50 | wmAP@50 | wmR@50 | Score@50 |
> | :--- | :--- | :--- | :--- | :--- | :--- | :--- |
> | Motifs | 71.33 | 31.82 | 51.575 | 29.86 | 31.52 | 38.154 |
> | Motifs + $RPC_{Rep}$ | 71.96 | 32.78 | 52.370 | 28.25 | 30.64 | 36.992 |
> | Motifs + $RPC_{Add}$ | 74.65 | 35.01 | 54.830 | 32.76 | 33.83 | 41.138 |
> | VCTree | 73.84 | 33.91 | 53.875 | 32.57 | 33.06 | 40.824 |
> | VCTree + $RPC_{Rep}$ | 74.10 | 34.29 | 54.195 | 31.44 | 32.56 | 39.972 |
> | VCTree + $RPC_{Add}$ | 75.99 | 37.65 | 56.820 | 34.60 | 35.42 | 42.878 |
> | BGNN | 75.30 | 39.80 | 57.550 | 32.90 | 34.50 | 41.380 |
> | ST-SGG | 75.20 | 42.90 | 59.050 | 32.80 | 34.20 | 41.280 |
> | RelDN | 75.30 | 37.20 | 56.250 | 32.20 | 33.40 | 40.820 |
> | GPS-NET | 74.81 | 35.26 | 55.035 | 32.85 | 33.98 | 41.242 |
> | HP | 76.34 | 41.46 | 58.900 | 34.81 | 35.42 | 43.116 |
> | HP-i | 76.50 | 37.97 | 57.235 | 38.44 | 39.29 | 46.052 |
>
> **Q2: Formatting Issue**
>
> **R2:**
> We have already noticed this issue and will correct the running header in the camera-ready version.
> Thanks for your time and your recognition of our work. We hope our response addresses your concerns. If you have any further questions, we would be more than happy to answer them.
>
> **Limitation**
>
> **R3:**
> Regarding limitations, RPC faces two main constraints:
>
> (1) Our plug-and-play module cannot alter the inherent unidirectional bias within the datasets. Relying solely on augmented inverse relations cannot entirely eradicate this issue at the model level. Fundamentally resolving this would require dataset optimization, such as annotating the missing inverse relations.
>
> (2) Since we do not modify the upstream object detector, our module is inevitably bottlenecked by the underlying baselines. For instance, some mispredictions from vanilla Motifs persist in Motifs+RPC—though RPC significantly improves the recognition of their inverse counterparts, as demonstrated in Table 2.
>
> Additionally, RPC yields marginal improvements on strictly symmetric relations (e.g., "and"). Nevertheless, it is foreseeable that employing stronger baselines in the future will naturally yield better overall performance.
>
> Thank you for your time! If our response has resolved your concerns, we would appreciate a higher score.

---

> > ### Author Rebuttal · Reviewer_Hgck · 2026-04-03
> >
> > thanks for the reply. i'll keep my positive score.

---

> > > ### Author Response · Authors · 2026-04-04
> > >
> > > We sincerely thank the reviewer for the valuable feedback and for maintaining the positive score. We truly appreciate your support for our work.
> > >
> > > We realize that our previous round of responses might not have been comprehensive enough to resolve all your concerns. Since specific remaining issues were not detailed, we completely understand that certain complex aspects of our framework might require a more in-depth discussion than a short rebuttal allows. We would like to take this opportunity to further clarify the following details regarding the datasets and metrics:
> > >
> > > Compared to Visual Genome (VG), the OpenImages (OI) dataset contains a larger number of entities but relatively fewer relations. The key reason is that OI's annotations are extremely sparse. While this mitigates data annotation bias to some extent, it also inherently overlooks the unidirectional bias that our work addresses. As shown in our tables, although OI retains the traditional R@K and mR@K metrics, it fundamentally places a much stronger emphasis on precision. Because the score is heavily dominated (80%) by precision, we primarily utilized the VG dataset to emphasize our MPIR framework and its associated new metrics, IR@K and ImR@K.
> > >
> > > This metric bias also explains the performance differences of our RPC method on the OI dataset. Although the "Rep" (Replace) mode improves Recall metrics, its performance in precision (and thus the final score) is less optimal. This is because the Rep strategy replaces some original relations with inverse relation data samples during training. Even though this strategy effectively helps disambiguate the model's unidirectional bias, it gets naturally penalized by OI's precision-weighted formula. Conversely, under the "Add" mode, our method shows significant improvements across the board, which further demonstrates our framework's effectiveness in mitigating unidirectional bias.
> > >
> > > We are committed to incorporating all your constructive suggestions to update and polish the final version of the paper. Furthermore, we will detail the newly added datasets and related works (including inverse relation generation and entity category definitions) in the appendix.
> > >
> > > We sincerely hope these detailed explanations can address your core concerns and earn your complete approval. If we missed any specific details, please feel free to let us know. Thank you again for your time and guidance!

---

### Official Review · Reviewer_WNE8 · 2026-03-13

**Soundness:** 3
**Presentation:** 2
**Significance:** 3
**Originality:** 3
**Overall Recommendation:** 4
**Confidence:** 4

**Summary:**

The paper proposed the RPC framework (Reciprocal Perspective Calibration), a model-agnostic approach, to address the unidirectional bias. The RPC framework is a plug-and-play method to establish MPIR, the logical inverse of predicted actions in the scene graph, for eliminating the violation of physical reality and diminishing the violation of self-inconsistency. The RPC either augmented the dataset or replaced training data, depending on the predicate's categories, to enrich the training dataset's distribution.
The authors also propose the Hypernym-Guide prompting (HGP) technique to enable CLIP to perform SGG with computational efficiency.
The results show that injecting the RPC and HPG further improves the SOTA model's performance on the scece graph generation task.

**Compliance With Llm Reviewing Policy:**

Affirmed.

**Final Justification:**

The concerns regarding the clarification and limitation of the paper's framework are addressed, as are other concerns. I would increase my score toward accepantance of the paper.

**Key Questions For Authors:**

1. The selection of LLM for HGP does not seem to have much difference across the models, while small models like Qwen2.5 should have significant differences compared to GPT4o. What is the reason behind this?

2. The low and high frequencies discussed in Section 4.1.1 seem not to have been discussed before in Section 3.2.3, which explains the Type I and Type II. Are these Type I and Type II, and the same as mentioned in both sections?

There is no additional question; please address the weaknesses raised above.

**Limitations:**

Yes

**Strengths And Weaknesses:**

# Strengths

- The paper proposed a plug-and-play framework that can be applied to SOTA models to further improve the performance on SGG.

- The results illustrate the improvement over several injected baselines, illustrating the effectiveness of the method across different model types.

- The paper provides the details of motivation and basic principles to understand the proposed concepts.

- An ablation study is shown to ensure the essential elements of each proposed component.

---

# Weaknesses

- The methodology sections seem to be challenging to understand in some parts. Mentioning figures 3 and using examples in the figures to help explain this section should make it easier to understand.

- There is no analysis of a failure case in which the proposed method cannot be solved. This could be used to strengthen the paper and guide future tracing for improvement.

- The empirical computation overhead of the proposed method is not discussed. Some parts of the proposed method seem to increase or decrease computational complexity compared to the traditional baseline. Augmenting the dataset using RPC may also need to be discussed in terms of the computational overhead during training.

---

> ### Author Rebuttal · Authors · 2026-03-31
>
> **R1 for W1**
>
> To clarify our methodology, we briefly walk through Section 4 alongside Figure 3. Our core innovation, the MPIR principle (formalized in Sec. 3), resolves unidirectional bias, while Sec. 4 introduces the built-upon RPC framework:
>
> 1. RPC Framework & AIRA (Fig. 3, bottom): An input image yields a standard subject-anchored triplet (left). Guided by MPIR, shifting the anchor (swapping subject/object) derives the inverse triplet. The right side details the AIRA strategy (Sec. 4.1.1), deploying differential treatments: Additive Injection for Semantic/Tail classes, and Stochastic Replacement for Geometric/Head classes. This mitigates unidirectional bias without exacerbating long-tail or information imbalances. Sec. 4.1.2 then details RPC as a plug-and-play module for open/closed-set constraints.
>
> 2. Overall Pipeline & HGP (Fig. 3, top): The top-left depicts HGP (Sec. 4.2). Unlike prior methods using fixed templates or omitting entities in dynamic prompts, HGP embeds coarse-grained semantic hypernyms alongside learnable contexts. This preserves crucial subject-object interactions while avoiding combinatorial explosion in the prompt space. The constructed textual prompts seamlessly align with baseline pipelines, enabling inverse relation recognition.
>
> 3. Knowledge Distillation (Sec. 4.3): To formalize the inverse mapping (Φ) and hypernym projection (Ψ) functions, we distill knowledge from LLMs. Refined via human prompts and empirical calibration, this knowledge base replaces rigid heuristics, ensuring robust scalability and laying a flexible foundation for future MPIR-based explorations.
>
> We will explicitly integrate these figure-guided explanations into the revised methodology section.
>
> **R2 for W2**
>
> Failure Cases: Due to space constraints, please refer to our response to Reviewer Hgck.
> Future Work: We will continuously explore MPIR to resolve unidirectional bias and unrecognized inverse relations. We anticipate that our data augmentation strategy will significantly benefit semi-supervised learning and Open-Vocabulary SGG. Specifically, inverse relations can serve as abundant, low-cost pseudo-labels or rare relation samples to boost open-vocabulary capabilities. Additionally, we will leverage our proposed metrics to further optimize inverse relation recognition.
>
> **R3 for W3**
>
> As a plug-and-play module, RPC introduces minimal training time overhead. Within AIRA, the Rep strategy solely replaces data, while Add only augments scarce tail data, strictly limiting dataset expansion. During inference, RPC achieves zero extra computational overhead.
>
> Empirical details substantiate this:
>
>     Memory Footprint: The peak memory of the baseline is exactly 14301 MB. With RPC and inverse augmentation, the peak remains precisely 14301 MB, indicating absolutely no additional memory burden.
>
>     Time Overhead: The baseline's average iteration time is 0.626–0.635s. During stable runs, RPC introduces merely a 3–4% time increase, which is entirely acceptable given the significant gains.
>
>     HGP Efficiency: As analyzed in Sec. 4.2, brute-force full-triplet enumeration for prompts triggers combinatorial explosion. In experiments, this causes severe GPU Out-of-Memory (OOM) errors, rendering the model unrunnable. HGP effectively averts this.
>
> **R4 for Q1**
>
> As detailed in Appendix C, we guided the LLMs using simple constraints and few-shot examples, effectively combining human priors with the model's capabilities. This implies that the task only requires fundamental linguistic competence to yield the desired outputs. Consequently, regardless of parameter size, almost any standard LLM can reliably generate similar, commonsense-aligned results. This robustness across different models actually corroborates the high reproducibility of our framework. Furthermore, because LLMs are inherently prone to hallucinations and potential fallacies, we do not rely on them blindly; integrating human intervention with the LLM's raw output ultimately yields the optimal and rigorous results.
>
> **R5 for Q2**
>
> Regarding the distinction between frequency and predicate types, we would like to clarify that "low/high frequencies" and "Type I/II" are fundamentally distinct concepts and do not possess a strict equivalence. We have meticulously maintained the consistent definitions of Type I and Type II throughout the entire manuscript.
> Specifically, low and high frequencies describe the empirical occurrence statistics of relations within the dataset, representing the tail and head of the long-tailed distribution, respectively. The detailed categorization of these frequency-based splits is provided in Appendix A.4.
> Conversely, Type I and Type II, as formally defined in Section 3.2.3, denote the intrinsic logical and physical properties of the relations themselves, strictly referring to Semantic Relations and Geometric Relations, respectively.
>
> Thank you for your time! If our response has resolved your concerns, we would appreciate a higher score.

---

> > ### Author Rebuttal · Reviewer_WNE8 · 2026-04-04
> >
> > Thanks for the author's responses. The concerns regarding the clarification and limitation of the paper's framework are addressed, as are other concerns. I would increase my score toward accepantance of the paper.

---

> > > ### Author Response · Authors · 2026-04-04
> > >
> > > We sincerely thank the reviewer for carefully reading our rebuttal and for the positive feedback. We are glad that our additional clarifications regarding the framework and limitations have fully resolved your concerns. Your constructive suggestions have been invaluable in helping us improve the paper. We deeply appreciate your time, effort, and support for our work!

---

### Official Review · Reviewer_t9jc · 2026-03-13

**Soundness:** 2
**Presentation:** 3
**Significance:** 2
**Originality:** 2
**Overall Recommendation:** 4
**Confidence:** 3

**Summary:**

This paper tackles the unidirectional bias in scene graph generation framework, which originates from lack of explicit handling of inverse relations, and proposes mutual perspective inverse relations (MPIR) principle and reciprocal perspective calibration (RPC) framework. Specifically, the authors analyzed different types of predicates and generated inverted triplets based on those analysis, and proposed Hypernym-Guided Prompts (HGP) to optimize the prompt to accurately capture the inverted predicates. The authors verifies that the proposed method improves existing SGG methods in PredCl tasks in Visual Genome dataset.

**Compliance With Llm Reviewing Policy:**

Affirmed.

**Final Justification:**

Clarifications during the rebuttal period resolves my main concerns regarding the motivation of this paper and why this method is actually helpful. The limitations of post‑hoc approaches and the importance of addressing unidirectional bias of the model are clearly clarified.
Therefore, I decided to raise my score to weak accept.

**Key Questions For Authors:**

- Can the authors elaborate more on why explicit modeling of inverse relations is necessary? The detailed examples and limitations in practical applications would be helpful.

- In HGP, it is not entirely clear how prompt optimization on context vectors $\mathbf{U}$ is conducted and on which objective. Although Algorithm 1 in the Appendix contain detailed procedure, it is not referenced in the main paper and some of the lines are not clear (It is written that “Compute probability P(p|s,o) via Eq. (5)” but Eq. (5) just denotes HGP embedding.) To make it self-contained, it is encouraged to add more details in Section 4.2.3.

**Limitations:**

No, it is encouraged to discuss failure cases that are incorrectly handled by RPC.

**Strengths And Weaknesses:**

**Strengths**
- The paper is well organized and easy to follow.
- Tackling an unidirectional bias in SSG is an interesting direction.
- The proposed method can work on top of existing SSG without modification.

**Weaknesses**
- **The motivation of this work is not entirely convincing**: While tackling an unidirectional bias in SSG is interesting and a consistency property can be beneficial, it is not entirely convincing that this problem is actually a critical problem. In particular, it remains unclear why a simple post-hoc method such as using an LLM to generate inverse triplets from predicted triplets in SSG could not handle the unidirectional bias problem. In other words, if we already know the triplet $<s,p,o>$, it is not clear which kinds of practical problems are expected when we do not explicitly having $<o, p^{-1}, s>$. It seems that we can handle it by simply paraphrasing the predicate.

- **The benefit of downstream utility is not empirically supported.**: Related to the first concern, the authors motivate handling inverse relations by arguing the benefit of downstream tasks such as VQA and image retrieval in Section 3.1, but it is not currently supported by the empirical evidence. The experiments are conducted only on VG PredCl tasks, but it only assesses how it predicts accurate predicates given subject and object. To convince the importance of explicitly modeling inverse relation and support the claim stated in Section 3.1, evaluation on other downstream tasks such as VQA or Image Retrieval (examples provided by the authors) should be provided.

- **The design of the proposed method is somewhat heuristic and not scalable:** In Section 3, the authors defined three different types (Type1,2,3) of predicates and provided corresponding inverse generation rules, and it remains unclear why this hand-guided design is necessary. In other words, could we just prompt LLM to find  $<o, p^{-1}, s>$ from $<s,p,o>$? While such a naive method would be unstructured and open-set prediction, it would be very simple, scalable and flexible as LLM would adaptively find proper inverse predicate. If the role of such hand-designed rules is to match the closed-set SSG,
Section 4.1.2 already filters OOD predicates by keeping only $p^{-1}\in \mathcal{C}$, then
using LLM for open-set prediction would not cause any issues. The reviewer may not correctly understand the authors’ intention, so it would be appreciated if the authors clarify on this design choice.


- **Closed-set filtering degrades the mean Recall**: In Table 1, the result of closed-set filtering leads to a significant drop in mean Recall and the performance is improved only when prompt optimization is applied for open-set settings.
Then, could we just use LLM to directly generate an inverse predicate $p^{-1}$? It would be the simple and scalable way to get inverse predicates without hand-designed rules.



- (Typo) In Figure 3, it seems the location of Text Encoder and Image Encoder are swapped.

---

> ### Author Rebuttal · Authors · 2026-03-31
>
> **W1:Motivation&post-hoc**
> R1:First, our motivation (Sec 3.1) stems from the severe unidirectional bias in SGG: relation prediction collapses when the observation anchor shifts (model-side), and inverse relation annotations are missing (dataset-side). Second, the suggested post-hoc LLM generation faces several critical limitations: (1) Unconstrained inverse relation generation easily introduces ambiguity and factual errors. (2) While MPIR-based post-hoc paraphrasing can alleviate dataset bias (which corroborates MPIR's effectiveness and aligns with our core idea), it operates independently of the relation recognition stage. Therefore, it cannot optimize the model's inherent unidirectional bias. Post-hoc methods are better suited to downstream tasks, whereas we aim to fundamentally improve the SGG model itself. (3) Post-hoc methods heavily rely on the initial SGG predictions. Given existing SGG challenges like long-tail distributions, applying post-hoc inverse generation on incorrect predictions will only reinforce the model's false confidence. This exacerbates cognitive fallacies and further hinders downstream applications.
>
> **W2:Downstream Utility**
> R2: Downstream evaluations are promising and would likely further highlight MPIR's utility, but they are beyond this paper's scope. Nevertheless, we demonstrate MPIR’s relevance to VQA via fine-grained SGG empirical analysis.
> (1) VQA queries are highly dynamic and perspective-variant, agnostic to subject-object distinctions. Traditional SGG models favor salient objects and fail when the observation anchor shifts. As shown in Table 2, baselines collapse under such shifts, whereas our model remains robust.
> (2) VQA (GQA dataset) involves complex cross-object queries requiring precise Semantic and Possessive relation recognition. Our AIRA strategy (Sec. 4) directly addresses the long-tail distribution to meet these demands. Thus, MPIR (Sec. 3) explicitly corrects unidirectional bias to generate optimal scene graphs for downstream reasoning.
> (3) In future work, we will integrate inverse relations into downstream models and evaluate the viability of the post-hoc method on VQA tasks.
>
> **W3:Scalability**
> R3:(1) Rather than relying solely on manual rules, we regulate the LLM by injecting human priors into prompts. As detailed in Appendix C.2, RPC supports flexible and customizable relation generation with strong scalability. (2)The Type 1-3 classification reflects the intrinsic properties of predicates. Such prompt constraints effectively mitigate LLM hallucinations. Unconstrained LLMs tend to generate multiple ambiguous and misaligned inverse relations, paradoxically demanding heavier post-filtering. (3)Due to space constraints, please refer to our response to Reviewer Hgck for further LLM-related discussions.
>
> **W4:Table 1**
> R4:We apologize for the confusion over Table 1's layout. To clarify, the closed-set filtering results do not degrade performance. Our RPC framework is actually applied to the Motifs and VCTree baselines (denoted as Motifs+ and VCTree+ in our analysis). MLLMP is included solely as a recent SOTA model for lateral comparison.
> Therefore, the "+RPCRep" and "+RPCAdd" rows should be grouped immediately below Motifs and VCTree. When correctly compared against their actual baselines, the addition of RPC yields solid improvements in mean Recall (mR@K). We will reorganize the table to prevent further misunderstanding, though the empirical results perfectly align with our textual descriptions.
>
> **Typo:** Thanks! We will correct it in the final version.
>
> **Q1: Inverse Relations**
> R5:As detailed in Sec. 3.1, explicit modeling of inverse relations is crucial for the model to genuinely comprehend relationships. It prevents subject-object confusion when the observation anchor shifts. For a practical example (see Fig. 1), simply swapping the subject and object causes the baseline to absurdly predict "beach walking on man." Moreover, our empirical analysis (particularly Table 2) highlights a severe limitation in applications: existing models perform exceptionally poorly on inverse relation prediction.
>
> **Q2:HGP Optimization**
> R6:(1) As a plug-and-play module detailed in Sec. 4.2 (illustrated above the RPC in Fig. 3), HGP employs simple prompt templates. Specifically, we insert learnable context vectors into the prefix and integrate the subject and object as hypernyms. For a more comprehensive explanation, please refer to our response to Reviewer WNE8.
> (2) We apologize for the typo regarding the formula references. The probability computation should actually reference Eq. (10), and the optimization objective is Eq. (8), detailed in Appendix A.1. We will correct these errors and add the requested details to Sec. 4.2.3 to make it self-contained in the revised version.
>
> **Limitation:**
> We sincerely apologize that our rebuttal is constrained by the strict word limit. Please refer to our response to Reviewer Hgck.
> Thank you for your time & suggestions for improvement.

---

> > ### Author Rebuttal · Reviewer_t9jc · 2026-04-04
> >
> > I thank the authors for the detailed explanations, which addresses my concern regarding the motivation by clarifying the limitations of post-hoc approaches and the need to correct unidirectional bias at the model level.
> > Therefore, I decided to raise my score.

---

> > > ### Author Response · Authors · 2026-04-04
> > >
> > > We really thank the reviewer for dedicating time to read our rebuttal and for the encouraging decision to raise the score.
> > >
> > > We are thrilled that our clarifications have resolved your concerns. It is deeply rewarding to know that you recognize the value and potential of our work. We strongly believe that introducing this new perspective to tackle unidirectional bias is a meaningful step forward, and your insightful feedback has significantly helped us refine the presentation of these core ideas.
> > >
> > > We truly appreciate your strong support and invaluable guidance throughout this review process!

---

### Decision · Program_Chairs · 2026-04-30

**Decision:**

Accept (regular)

**Comment:**

The paper proposes a Reciprocal Perspective Calibration (RPC) framework to address unidirectional bias in scene graph generation (SGG) by explicitly modeling inverse relations through the Mutual-Perspective Inverse Relations (MPIR) principle and a prompt-based augmentation strategy. The reviewers highlighted several strengths of the work, including (1) a clear and novel problem formulation, (2) a model-agnostic, plug-and-play framework, with a practical design combining data augmentation and prompt learning, (3) a conceptually grounded approach leveraging reciprocal reasoning and logical consistency, and (4) consistent empirical improvements across multiple baselines, supported by ablation studies.

However, reviewers also raised several concerns: (1) motivation and necessity of explicit inverse modeling compared to simpler post-hoc approaches; (2) limited empirical validation, including evaluation restricted mainly to PredCls on Visual Genome and lack of downstream task validation; (3) methodological complexity and scalability, including reliance on heuristic predicate categorization and LLM-based components; and (4) insufficient analysis, such as lack of failure cases, sensitivity to hyperparameters, and robustness to LLM noise or hallucination.

During the rebuttal discussion, the authors resolved most of the reviewers’ concerns: 1) they clarified the limitations of post-hoc inverse generation and emphasized the importance of correcting bias at the model level rather than at inference time; 2) they provided additional analysis on long-tail behavior, clarified the role and robustness of LLM-based prompting, and addressed concerns on efficiency, scalability, and hyperparameter sensitivity; and 3) they also extended evaluation to additional datasets (e.g., Open Images) and improved methodological clarity. After the rebuttal, all reviewers maintained or strengthened their positive assessments, with several explicitly raising their scores and indicating that key concerns had been addressed.

The AC concurs with the reviewers: while further validation on broader tasks and deeper analysis would strengthen the work, the paper presents a well-motivated and practically useful contribution to SGG by introducing reciprocal reasoning into relation modeling. Therefore, the AC recommends accept, and encourages the authors to further incorporate the rebuttal discussion in the final version.